# The phosphorylated pathway of serine biosynthesis affects sperm, embryo, and sporophyte development, and metabolism in *Marchantia polymorpha*

Mengyao Wang[1,2], Hiromitsu Tabeta [1,3,4,12], Kinuka Ohtaka[1,2,10,12], Ayuko Kuwahara[1], Ryuichi Nishihama [5,11], Toshiki Ishikawa [6], Kiminori Toyooka [1], Mayuko Sato [1], Mayumi Wakazaki[1], Hiromichi Akashi[1], Hiroshi Tsugawa [1,7], Tsubasa Shoji[1], Yozo Okazaki[1,8], Keisuke Yoshida [9], Ryoichi Sato[1], Ali Ferjani [4], Takayuki Kohchi [5] & Masami Yokota Hirai [1,2✉]

Serine metabolism is involved in various biological processes. Here we investigate primary functions of the phosphorylated pathway of serine biosynthesis in a non-vascular plant *Marchantia polymorpha* by analyzing knockout mutants of Mp*PGDH* encoding 3-phosphoglycerate dehydrogenase in this pathway. Growth phenotypes indicate that serine from the phosphorylated pathway in the dark is crucial for thallus growth. Sperm development requires serine from the phosphorylated pathway, while egg formation does not. Functional Mp*PGDH* in the maternal genome is necessary for embryo and sporophyte development. Under high $CO_2$ where the glycolate pathway of serine biosynthesis is inhibited, suppressed thallus growth of the mutants is not fully recovered by exogenously-supplemented serine, suggesting the importance of serine homeostasis involving the phosphorylated and glycolate pathways. Metabolomic phenotypes indicate that the phosphorylated pathway mainly influences the tricarboxylic acid cycle, the amino acid and nucleotide metabolism, and lipid metabolism. These results indicate the importance of the phosphorylated pathway of serine biosynthesis in the dark, in the development of sperm, embryo, and sporophyte, and metabolism in *M. polymorpha*.

[1] RIKEN Center for Sustainable Resource Science, Yokohama, Japan. [2] Graduate School of Bioagricultural Sciences, Nagoya University, Nagoya, Japan. [3] Graduate School of Arts and Sciences, The University of Tokyo, Tokyo, Japan. [4] Department of Biology, Tokyo Gakugei University, Tokyo, Japan. [5] Graduate School of Biostudies, Kyoto University, Kyoto, Japan. [6] Graduate School of Science and Engineering, Saitama University, Saitama, Japan. [7] Department of Biotechnology and Life Science, Tokyo University of Agriculture and Technology, Tokyo, Japan. [8] Graduate School of Bioresource, Mie University, Tsushi, Japan. [9] Institute of Innovative Research, Tokyo Institute of Technology, Yokohama, Japan. [10] Present address: Department of Chemical and Biological Sciences, Japan Women's University, Tokyo, Japan. [11] Present address: Department of Applied Biological Science, Faculty of Science and Technology, Tokyo University of Science, Tokyo, Japan. [12] These authors contributed equally: Hiromitsu Tabeta, Kinuka Ohtaka. ✉email: masami.hirai@riken.jp

Serine, an amino acid highly accumulated in plants[1], is involved in cell signaling in response to various environmental stresses and the biosynthesis of various biomolecules, such as nitrogenous bases, phospholipids, and sphingolipids[2–4]. In plants, three serine biosynthesis pathways exist and function coordinately in the daytime. The major pathway during daytime is the photorespiratory glycolate pathway in photosynthetic tissues[5–7] (Supplementary Fig. 1). In this pathway, glycine decarboxylase complex and serine hydroxymethyltransferase (SHMT) convert glycine to serine[8,9]. In dark environments and non-photosynthetic tissues, two other pathways are responsible for serine synthesis, namely, the glycerate and phosphorylated pathways[9,10].

The glycerate pathway starts with the dephosphorylation of 3-phosphoglycerate (3-PGA) by 3-PGA phosphatase in the cytosol or peroxisomes, followed by a sequence of reactions catalyzed by glycerate dehydrogenase (GDH) and alanine: hydroxypyruvate (serine: pyruvate) aminotransferase[5,11–13]. On the other hand, the phosphorylated pathway from 3-PGA occurs in plastids and is catalyzed by 3-PGA dehydrogenase (PGDH), 3-phosphoserine aminotransferase (PSAT), and 3-phosphoserine phosphatase (PSP)[9,14,15].

The phosphorylated pathway is conserved in animals, plants, and bacteria[4,16]. This pathway in plants is the only serine source for specific cell types and is essential for the embryo, pollen, male gametophyte, and postembryonic root development in *Arabidopsis thaliana*[3,17–19]. The phosphorylated pathway of serine biosynthesis (PPSB) has been suggested to play a fundamental role in plant responses to various environmental stresses, such as low temperature, high salinity, and pathogens[3,4,17,20]. Although the reason why plants take three routes for serine biosynthesis is unclear, serine production through the phosphorylated pathway is crucial in plant metabolism and development[3,21]. The genes encoding PGDH are important checkpoints in PPSB as they are under tight transcriptional control and responsible for several physiological events[3]. In *A. thaliana*, three PGDH isoforms (AtPGDH1, AtPGDH2, and AtPGDH3) are expressed in different organs/tissues and have different physiological functions. AtPGDH1 is the essential gene and its silencing causes developmental arrest in roots, embryos, pollen, and male gametophytes. AtPGDH2 has a partially redundant role with AtPGDH1, while AtPGDH3 seems to have additional functions unrelated to serine synthesis[17–19,22,23]. In rice (*Oryza sativa*), three OsPGDH genes were expressed in all tissues and development stages, and their expressions responded to abiotic stresses[24]. Additionally, four PpPGDHs and two AmtriPGDHs were identified in moss *Physcomitrium patens* and basal angiosperm *Amborella trichopoda*, respectively[25]. The biochemical properties of various PGDH enzymes are different among isozymes in terms of allosteric regulation by amino acids[25,26]. Considering that all land plant species examined, except *Marchantia polymorpha* L., possess different types of PGDH isozymes, duplication and functional diversification of the *PGDH* genes was necessary for the evolution of land plants to adequately control the serine supply[25,27].

*Marchantia polymorpha* is a model bryophyte species and a dioecious plant with a haploid genome. *M. polymorpha* is increasingly used as a model plant for physiological, genetic, epigenetic, metabolic, and evolutionary studies. Recent studies have clarified metabolic responses to wounding stress[28], cellular expansion and integrity[29], reproductive development[30], as well as the mechanisms of spermatogenesis and embryogenesis[31–35]. Our previous study characterized the sole PGDH enzyme, MpPGDH, a 65.6 KDa protein sharing 75–80% identity with AtPGDHs[27]. MpPGDH has similar biochemical characteristics to AtPGDH1 in vitro, such as cooperative inhibition by L-serine and activation by

L-alanine, L-valine, L-methionine, L-homoserine, and L-homocysteine[27].

In this study, we aim to clarify the in vivo function of MpPGDH and explore the specific functions of serine from the phosphorylated pathway in a non-vascular plant *M. polymorpha*. Our results revealed the particular importance of serine from the phosphorylated pathway on vegetative growth, male gametogenesis, and sporophyte development in *M. polymorpha* and that PPSB is involved in the metabolism of the tricarboxylic acid (TCA) cycle, amino acids, nucleotides, and lipids, in a different manner under different light and $CO_2$ conditions in *M. polymorpha*. This study proposes that serine homeostasis is a key factor for robustness of not only metabolism but also growth and development.

## Results

**Serine supply from the phosphorylated pathway in the dark is crucial for thallus growth.** The expression pattern of MpPGDH was clarified using the *pro*MpPGDH:GUS transgenic lines. In vegetative phase, MpPGDH was expressed in almost the entire gemma (Supplementary Fig. 2a) and displayed a linear expression pattern in the midribs from the center of the thallus to the apical notches in 1- to 4-week-old thalli (Supplementary Fig. 2b–d).

The knockout mutants of MpPGDH were generated by introducing the CRISPR/Cas9 constructs into sporelings (Supplementary Fig. 3a). The reduced transcript level of MpPGDH and no detectable level of MpPGDH protein were confirmed in both male and female Mppgdh mutants (Supplementary Fig. 3b, c).

Considering that photosynthesis may affect serine biosynthesis in plants, thalli were grown under 16-h light/8-h dark (L/D) and continuous light (CL) conditions for comparison. Under the L/D condition, male Mppgdh-1 and Mppgdh-2 lines were significantly small in thallus size and fresh weight (Fig. 1) compared with the wild type. However, the severe growth phenotype of Mppgdh mutants was resolved by the exogenously-supplemented serine (Fig. 1). In contrast, no difference was observed between the mutants and wild type under the CL condition, while *M. polymorpha* grew better under this condition than the L/D condition (Fig. 1). Similar growth phenotypes were observed in the female Mppgdh knockout mutants (Mppgdh-3, Mppgdh-4, and Mppgdh-5; Supplementary Fig. 4). For further experiments, Mppgdh-1 and Mppgdh-3 were used since the supply of serine restored the growth of these mutants to the wild-type levels.

Poor growth phenotype observed only under L/D conditions suggested the importance of the PPSB in the dark. Then, the transcript levels of some key genes involved in three serine biosynthesis pathways were analyzed in the thallus transferred from CL to dark conditions. The expression of MpPGDH and MpPSAT in the PPSB was significantly induced at 8 and 16 h post transfer to darkness, respectively, compared with CL conditions (Supplementary Fig. 5). The MpGDH and MpSHMT expression levels in the glycerate and glycolate pathways, respectively, were gradually reduced in darkness. These results suggest that the PPSB was enhanced and functioned as the primary serine synthesis pathway under dark conditions.

To determine whether poor growth of the mutant thalli under L/D conditions was attributed to insufficient serine content, free amino acid contents in the thallus were measured. The serine content in Mppgdh mutants was significantly decreased after 4 h in the dark and restored to wild type-level after 2 h in the light (Fig. 2). These results suggested that reduced serine content during the dark period caused the poor growth in Mppgdh mutants under L/D conditions.

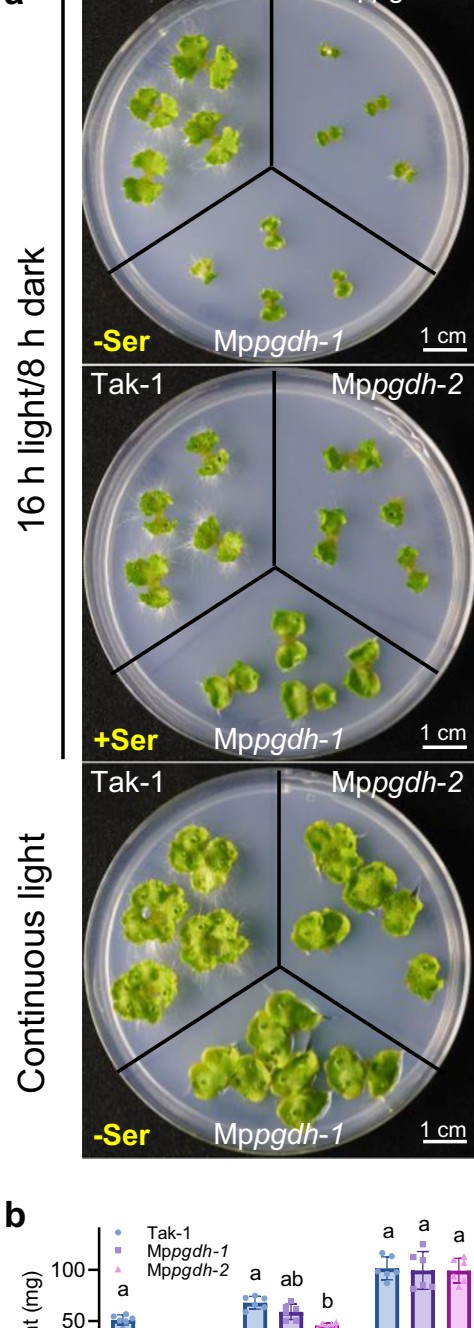

**Fig. 1 Thallus growth in the male Mp*pgdh* mutants. a** Plants grown on ½ B5 agar medium for 14 days with or without serine supplementation under 16 h light/8 h dark (L/D) or continuous light (CL) conditions. **b** The fresh weight of Mp*pgdh-1*, Mp*pgdh-2*, and wild-type Tak-1. Data represent means ± SD of six biological replicates (*n* = 6). One-way ANOVA followed by Tukey's test (*p* < 0.01) was performed in each group; columns with the same letter are not significantly different.

Overall, these results indicate that the perturbation of the PPSB impaired thallus growth under the L/D condition, revealing the crucial function of PPSB in the dark for thallus growth.

**MpPGDH-mediated serine synthesis is essential to sperm development.** The growth phenotype of Mp*pgdh* mutants was observed in the whole life cycle under the L/D condition to understand the importance of PPSB. The formation of gemma cups and gemmae in the Mp*pgdh* mutants was normal, likewise those of wild types (Supplementary Fig. 6).

The male Mp*pgdh* mutants developed umbrellalike antheridiophores (male reproductive branch) under far red-supplemented long-day conditions, which induce sexual reproduction in wild type (Fig. 3a, b). Antheridiophore development is generally divided into five stages (Supplementary Fig. 7a)[36]. When water was dropped on the dorsal surface of the antheridiophores at stage 4, white sperm masses discharged from antheridia emerged through the antheridial pores in wild type (Fig. 3c). Notably, no white sperm mass was observed in the Mp*pgdh* mutants after water application (Fig. 3c and Supplementary Fig. 7b). When cells in water were visualized via fluorescence staining of DNA, sperm with thin and crescent-shaped nucleus were visible, and very few sperm cells were observed in the Mp*pgdh-1* compared with wild-type Tak-1 (Fig. 3d).

To ascertain which process of sperm development (Fig. 3e) was impaired in the Mp*pgdh* mutants, the sections of resin-embedded antheridia were analyzed using field emission scanning electron microscopy (FE-SEM) at the early-, middle-, and mature stages. In wild-type Tak-1, spermatid mother cells were rectangular (Fig. 3f, top left) and underwent diagonal cell division to produce spermatids (Fig. 3f, top middle). In the spermatid stage, flagellar formation was observed. Lastly, the spermatids differentiated into sperm cells (Fig. 3f, top right). However, in the Mp*pgdh* mutants, spermatid mother cells were in various sizes and irregular shapes (Fig. 3f, bottom left), and subsequent diagonal cell division to generate spermatids was barely observed (Fig. 3f, bottom middle and right). The qRT-PCR analysis of sperm differentiation-related genes (Supplementary Fig. 8) revealed that the expression of Mp*PRM*, which is expressed specifically in sperm during late spermiogenesis[37], were significantly repressed. On the other hand, the expression level of Mp*DUO1*, which is an upstream regulator of Mp*PRM* and expressed in spermatid mother cells and spermatids[36], was similar in the wild type and the mutants. These results support the notion that cell division of spermatid mother cells to generate spermatids was blocked in the Mp*pgdh* mutants.

The GUS staining of the male *pro*Mp*PGDH:GUS* lines indicated that Mp*PGDH* was expressed in the middle area of antheridial receptacles (Supplementary Fig. 2e). Analysis using the Expression Database for *M. polymorpha*[37] confirmed that Mp*PGDH* is expressed in antheridia (Supplementary Fig. 2g).

To clarify whether exogenous serine supplementation and continuous light conditions rescue the defective spermatogenesis phenotype, the mutants grown under L/D + serine conditions and CL conditions were microscopically observed. Under these two conditions, the phenotype of Mp*pgdh* mutants was partially rescued (Supplementary Fig. 7c–f). Sperm masses were discharged in the Mp*pgdh* mutants (Supplementary Fig. 7c, e), but fewer sperm cells were observed compared with wild type (Supplementary Fig. 7d, f). The FE-SEM images indicated that the cell division of spermatid mother cells in Mp*pgdh-1* was rescued to some extent by serine supplement and CL, since diagonal cell divisions were visible in some regions in one antheridium (Supplementary Fig. 7g, h). The borders between the areas comprising cells in different states were clearly observed (Supplementary Fig. 7g, h), suggesting that

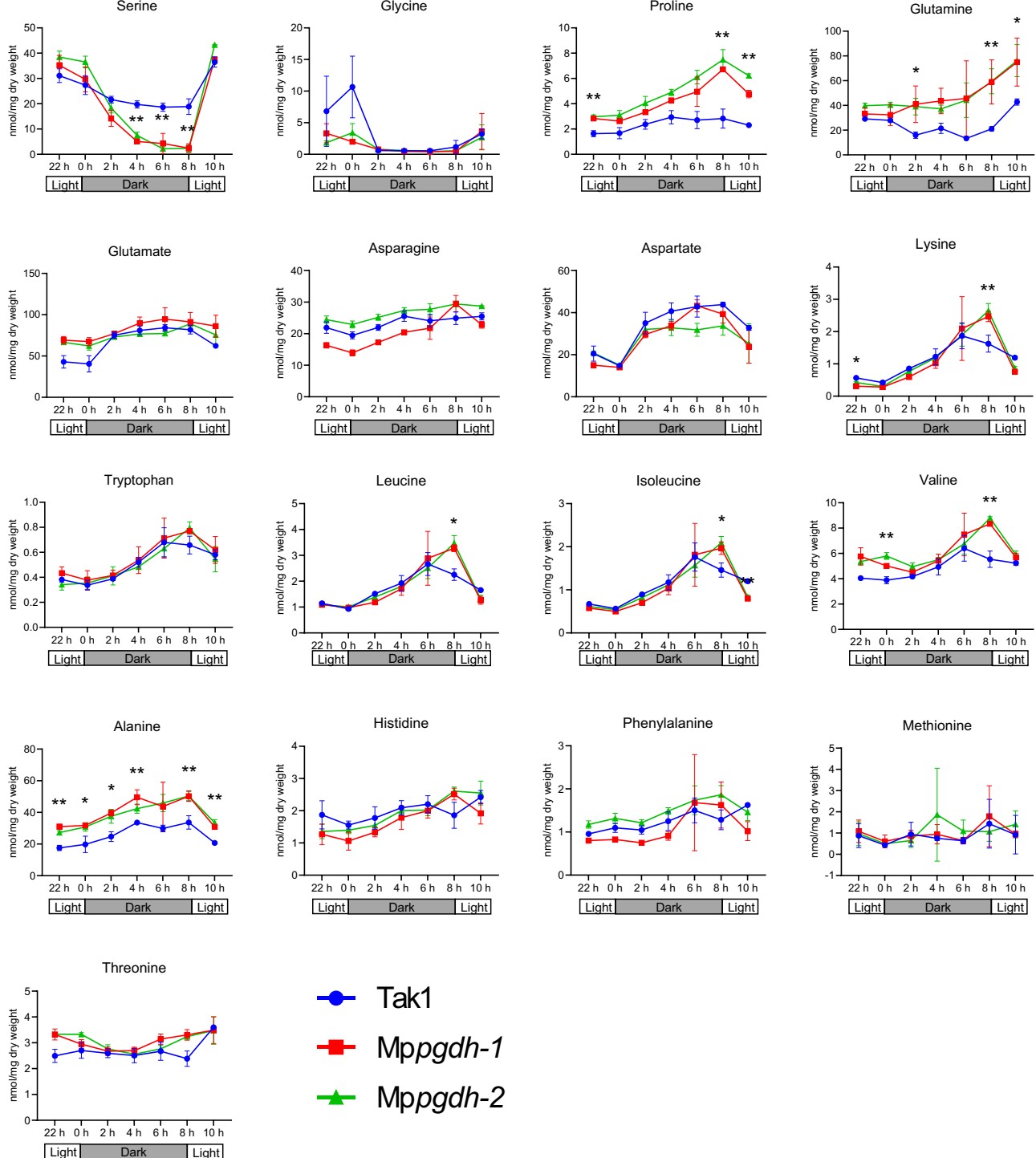

**Fig. 2 Free amino acid contents in thalli of the Mp*pgdh* mutants under L/D conditions.** The thalli of male lines grown under 16 h light/8 h dark conditions were sampled at 0 h, 2 h, 4 h, 6 h, and 8 h in the dark period, and at 2 h and 14 h in the light period. The free amino acid contents were measured using gas chromatography–quadrupole mass spectrometry. Data represent means ± SD of three biological replicates. Student's *t*-test was performed between wild type and one mutant respectively. Asterisks indicate statistically significant differences between the wild type and both two mutant lines (**p* < 0.05, ***p* < 0.01).

groups of neighboring cells in one antheridium share a similar status in terms of serine supply in Mp*pgdh-1*. Partial recovery by exogenous serine supplement and CL conditions may result in fewer number of sperm cells per antheridium in the mutants compared with the wild type.

In the female Mp*pgdh* mutants, no apparent morphological differences were observed in the formation of archegoniophores

(female reproductive branch, Fig. 4a, b) and archegonia (Fig. 4c) under the L/D condition supplemented with far-red light, compared with that in the wild type. Analysis using the Expression Database for *M. polymorpha* indicated that Mp*PGDH* is expressed in archegoniophore and archegonium (Supplementary Fig. 2g).

These results suggest that male gametogenesis required serine supply from the phosphorylated pathway.

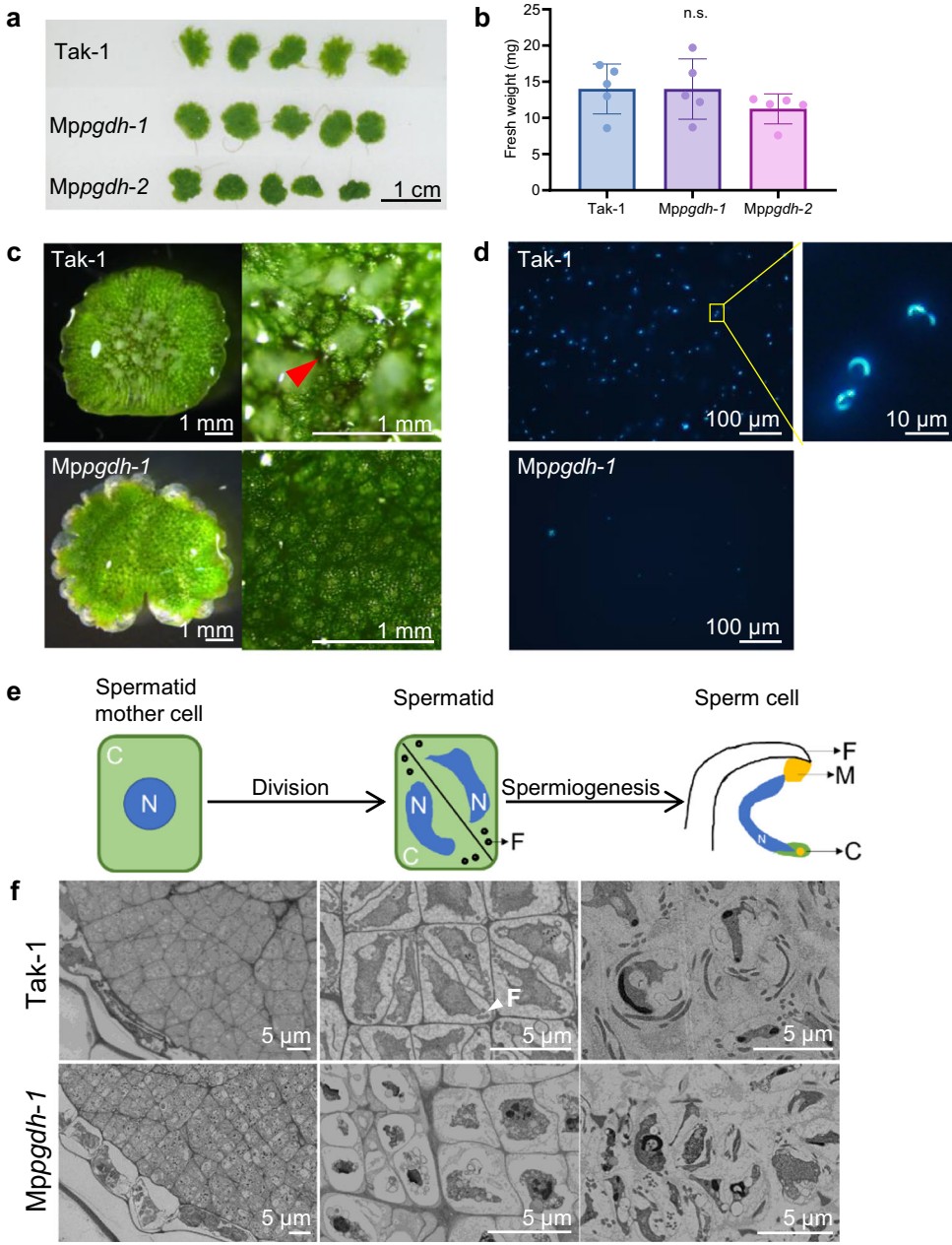

**Fig. 3 Male gametogenesis in the Mp*pgdh* mutants. a** Images of antheridial receptacles of Tak-1, Mp*pgdh-1*, and Mp*pgdh-2* grown under L/D conditions. **b** The fresh weight of antheridial receptacles in (**a**). Data represent means ± SD of five biological replicates ($n = 5$). One-way ANOVA followed by Tukey's test ($p < 0.05$) was performed (n.s., no significant difference). **c** Discharge of sperm masses. The white sperm masses (red arrowhead) were visible 10 min after dropping 50 µL water on the dorsal surface of antheridial receptacles. **d** Fluorescent staining of the cells. The cells in 10 µL water taken from (**c**) were visualized via Hoechst staining. Nuclei of sperm were very thin and crescent-shaped. **e** Schematic diagram of sperm development in wild type. **f** Field emission scanning electron microscopy (FE-SEM) images showing the process of *M. polymorpha* male gamete development. Cells in an early-stage antheridium (left), a middle-stage antheridium (middle), and a mature antheridium (right) are shown. The white arrowhead indicates the flagella. C cytoplasm, N nucleus, F flagella, M mitochondria.

**Knockout of Mp*PGDH* in maternal genome affects sporophyte development.** We observed the process on the archegoniophore of wild-type Tak-2 and Mp*pgdh*−3 after fertilization with sperm from wild-type Tak-1. In the Tak-2 x Tak-1 cross, yellow spores were generated and released from sporangia (Fig. 5a), whereas in the Mp*pgdh-3* x Tak-1 cross, only empty involucres were developed (Fig. 5b). Cross-sectional images of the developing sporophyte show the detailed process of embryogenesis and sporophyte development post fertilization (Fig. 5c–h). At 1-week post fertilization, a young sporophyte circularly surrounded by a calyptra and pseudoperianth was observed in both Tak-2 x Tak-1

(Fig. 5c) and Mp*pgdh-3* x Tak-1 (Fig. 5d), indicating that fertilization successfully occurred in Mp*pgdh-3* x Tak-1 as well. However, a clear delay in embryogenesis was observed in Mp*pgdh-3* x Tak-1. In Tak-2 x Tak-1, sporophytes detached into the inner space of the capsule, and differentiation along the apical-basal axis into foot, seta, and sporangium, where sporogenous cells began to differentiate, was clearly evident at 2-week post fertilization (Fig. 5e). At 3-week post fertilization, sporocytes underwent meiosis to produce spores in Tak-2 x Tak-1 (Fig. 5g). In contrast, sporophyte development seemed arrested before the stage of apical-basal differentiation in Mp*pgdh-3* x Tak-1 (Fig. 5f, h),

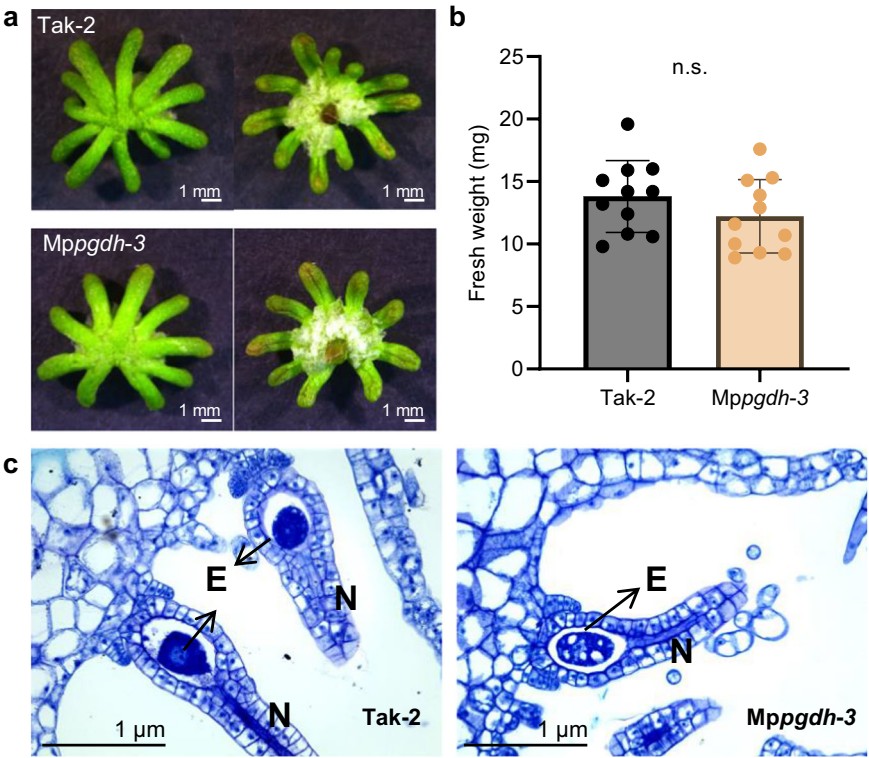

**Fig. 4 Oogenesis in the Mp*pgdh* mutants. a** Images of the archegonial receptacles of Tak-2 and Mp*pgdh-3* grown under L/D condition. **b** The fresh weight of archegonial receptacles of Tak-2 and Mp*pgdh-3*. Data represent means ± SD of 11 biological replicates ($n = 11$). Student's *t*-test was performed ($p < 0.05$, n.s., no significant difference). **c** The cross-section images of archegonia. E egg cell, N neck.

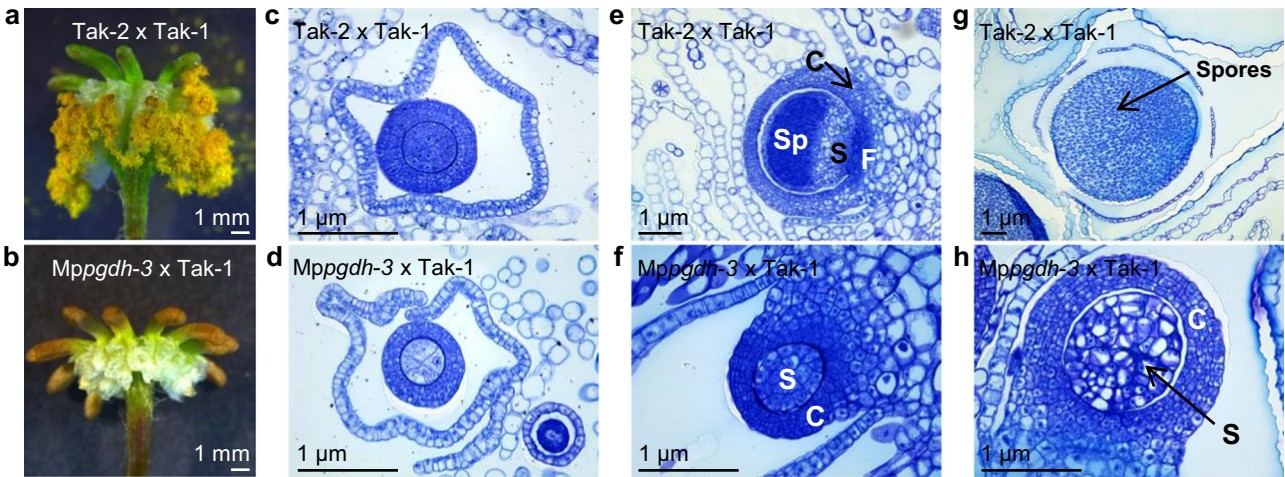

**Fig. 5 Sporulation in the female Mp*pgdh* mutant following fertilization with sperm from the wild type. a**, **b** The sporulation on Tak-2 (**a**) and Mp*pgdh-3* (**b**) approximately 1-month post fertilization with sperm from Tak-1. Images are representative of three archegoniophores. (**c**–**h**) The cross-section images of sporophytes at 1-week (**c**, **d**), 2-week (**e**, **f**), and 3-week (**g**, **h**) post fertilization. Tak-2 x Tak-1 (**c**, **e**, **g**), Mp*pgdh-3* x Tak-1 (**d**, **f**, **h**). C calyptra, F foot, S seta, Sp sporangium.

resulting in no spore production. Considering that paternal chromosomes are repressed during throughout embryogenesis (diploid phase)[35], this phenotype was attributed to the lack of the functional Mp*PGDH* in maternal genome. Interestingly, according to the Expression Database for *M. polymorpha*, the strongest expression of Mp*PGDH* was observed in sporophyte (Supplementary Fig. 2g). Under the CL conditions, the Mp*pgdh-3* x Tak-1 zygotes produced spore-bearing structures, and these spores germinated and grew into healthy individuals (Supplementary Fig. 9).

These results indicate that the knockout of Mp*PGDH* gene in maternal genome affected embryo and sporophyte development after fertilization.

**Mp*PGDH* knockout affects amino acid and nucleotide metabolism and the TCA cycle.** To determine how Mp*PGDH* knockout affects metabolism, we performed a widely targeted metabolome analysis.

The Mp*pgdh* mutants were grown under the L/D, CL, and L/D + serine conditions, and the metabolome in thalli was analyzed

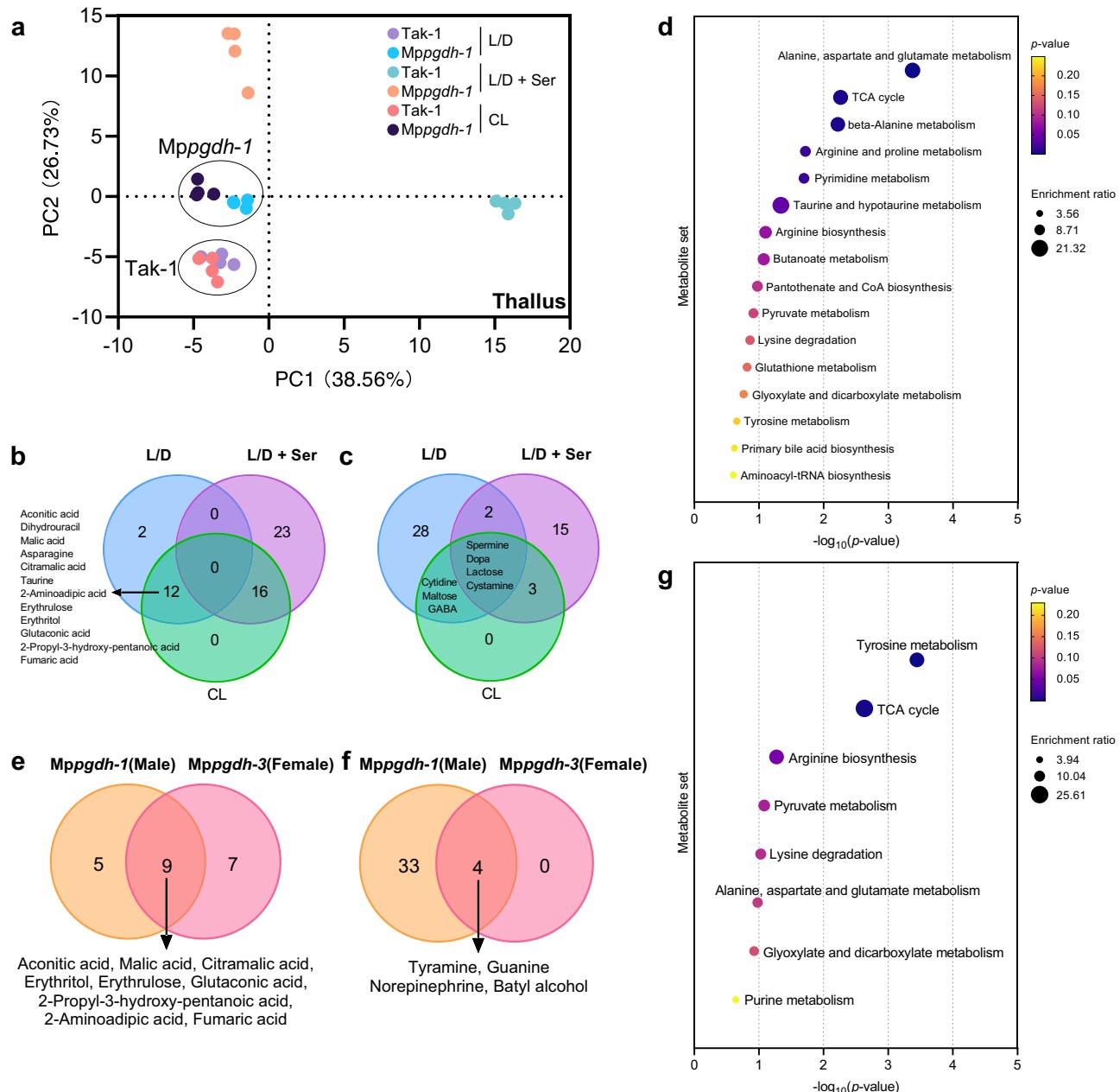

**Fig. 6 Changes in metabolome in 14-day-old thalli of the Mp*pgdh* mutants. a** PCA score plot of Tak-1 and Mp*pgdh-1* thallus samples grown under L/D, L/D + serine, and CL conditions ($n = 4$). **b**, **c** Venn diagrams showing the number of significantly decreased metabolites (**b**) and increased metabolites (**c**) in thalli of Mp*pgdh-1*. **d** KEGG pathway enrichment of common DAMs under L/D and CL conditions shown in (**b**, **c**). **e**, **f** Venn diagrams showing the number of significantly decreased metabolites (**e**) and increased metabolites (**f**) in thalli of Mp*pgdh-1* and Mp*pgdh-3* under L/D conditions. **g** KEGG pathway enrichment analysis of common DAMs shown in (**e**, **f**). In (**d**, **g**), vertical and horizontal axes indicate the metabolite set and the value of −log₁₀(*p*-value), respectively. The bubble size corresponds to the enrichment ratio. The color bar indicates the corrected *p*-value; yellow and navy blue represent higher and lower values, respectively. Dopa, 3,4-dihydroxyphenylalanine; GABA, γ-aminobutyric acid.

(Supplementary Fig. 10a). The principal component analysis (PCA) showed that the light condition did not greatly affect the metabolism of wide-type Tak-1 (Fig. 6a). In the PCA score plot, Mp*pgdh-1* formed the clusters separated from those of Tak-1 not only under L/D condition but also under CL condition, indicating that the PPSB also works at daytime. The metabolomes of Tak-1 and Mp*pgdh-1* supplemented with serine were clearly separated from the others, indicating that abundant serine supplement greatly affected the metabolism in the wild type and mutant in a different manner (Fig. 6a). Similarly, serine supplementation greatly affected the metabolome of the female lines, while the

wild-type Tak-2 and Mp*pgdh-3* were not separated in the PCA score plot under L/D and CL conditions without serine supplementation (Supplementary Fig. 11a). Thus, we mainly focused on the differentially accumulated metabolites (DAMs) under L/D and CL conditions in the following analyses. All the DAMs are listed in Supplementary Data 1.

The Venn diagrams (Fig. 6b, c) and volcano plots (Supplementary Fig. 10b, c) show that 12 DAMs were decreased in Mp*pgdh-1* commonly under L/D and CL conditions (Fig. 6b), while 7 DAMs were commonly increased under the two conditions (Fig. 6c). The Kyoto Encyclopedia of Genes and

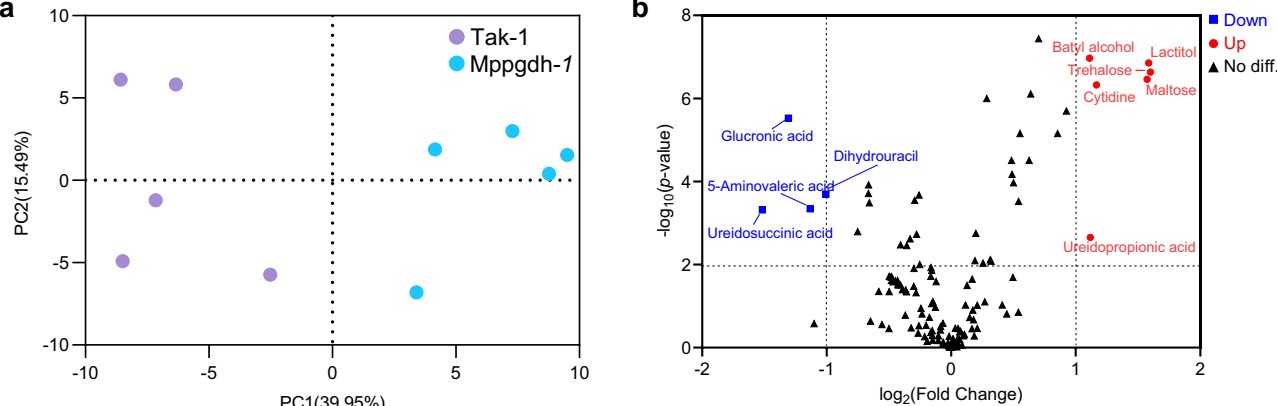

**Fig. 7 Changes in metabolome in antheridial receptacles of Mp*pgdh-1*. a** PCA score plot of the antheridial receptacle (stage 4) samples grown under L/D conditions ($n = 5$). **b** Volcano plot showing the DAMs in antheridial receptacles of Mp*pgdh-1*. Red dots and blue squares represent significantly increased ($p$-value < 0.01, fold change > 2) and decreased ($p$-value < 0.01, fold change < 0.5) metabolites, respectively, in Mp*pgdh-1*. Black triangles represent no significant differences between Tak-1 and Mp*pgdh-1*.

Genomes (KEGG) pathway analysis indicated that these DAMs were significantly enriched ($p < 0.01$) in "alanine, aspartate, and glutamate metabolism," "TCA cycle," and "beta-alanine metabolism" (Fig. 6d).

Similarly, DAMs in the female mutant Mp*pgdh-3* were analyzed (Supplementary Fig. 11b, c). Under the CL condition, metabolic profiles of Mp*pgdh-3* were restored nearly to wild-type level, although the serine content was still significantly reduced in Mp*pgdh-3* (Supplementary Fig. 11c). Focusing on common DAMs in Mp*pgdh-1* and Mp*pgdh-3* under L/D conditions, 9 metabolites decreased (Figs. 6e), and 4 metabolites (Fig. 6f) increased. These DAMs were enriched in "tyrosine metabolism" and "TCA cycle" (Fig. 6g).

We also analyzed metabolome of antheridial receptacles grown under L/D conditions, as Mp*pgdh-1* showed severe defect in spermatogenesis under this condition. The Mp*pgdh-1* and Tak-1 were separated in the PCA score plot (Fig. 7a), although metabolic profiles fluctuated slightly among replicate samples possibly due to the difficulty in clearly distinguishing antheridiophores at stages 3 or 4 when sampled (Supplementary Fig. 12a). The volcano plot (Fig. 7b) indicates 6 increased metabolites and 4 decreased metabolites, of which lactitol, trehalose, maltose, batyl alcohol, cytidine, and dihydrouracil were common DAMs in thalli and antheridial receptacles.

These results indicate that Mp*PGDH* knockout caused a metabolic change in the thallus and antheridiophore. Lack of functional Mp*PGDH* mainly affected the TCA cycle and amino acid metabolism. Metabolism was affected even under the CL condition, where the Mp*pgdh* mutants did not show a visible vegetative phenotype.

**Mp*PGDH* knockout alters lipid profiles.** Since serine is a precursor of various lipid species, we analyzed the lipidome in the Mp*pgdh* mutants.

The lipidome in thalli of the Mp*pgdh* mutants grown under the L/D, CL, and L/D + serine conditions were analyzed. In the PCA score plot (Fig. 8a, b), wild types and Mp*pgdh* mutants were clearly separated under L/D and CL conditions, indicating a large difference in lipid profiles. Like the metabolic profiles, exogenous serine supplements greatly influenced the lipidome (Fig. 8a, b). Focusing on the differentially accumulated lipid classes (DALCs) (Supplementary Fig. 13), the Venn diagrams showed that lysodiacylglyceryl trimethylhomoserine (LDGTS) and stigmasterol esters (STSE) decreased under CL conditions (Fig. 8c), while

sitosterol esters (SISE) increased under L/D conditions (Fig. 8d), commonly in male and female thalli. All the DALCs are listed in Supplementary Data 1.

Lipidome of antheridial receptacles grown under L/D conditions was also analyzed (Supplementary Fig. 12b). In the PCA score plot (Fig. 8e), Mp*pgdh-1* and Tak-1 were separated from each other. Triacylglycerols (TG), oxidized triglycerides (OxTG), and phosphatidylethanolamines (PE) were present in high amounts, whereas oxidized phosphatidylcholines (OxPC) was present in low amounts in Mp*pgdh-1* compared with that in Tak-1 (Fig. 8f).

These results suggest that Mp*PGDH* knockout greatly influenced lipid composition in vegetative and reproductive growth phases of *M. polymorpha*.

**PPSB is the primary serine synthesis pathway when photorespiration is inhibited.** To estimate the importance of the glycolate pathway on growth and metabolism, the wild types and mutants were grown under L/D conditions in high $CO_2$ (3000 ppm), which inhibited the glycolate pathway by suppressing photorespiration. GUS staining of *pro*Mp*PGDH:GUS* line showed a strong GUS signal in almost the entire thalli (Supplementary Fig. 2f), indicating that the Mp*PGDH* expression was induced under high $CO_2$.

Thalli growth of the mutant was significantly suppressed under high $CO_2$ (Fig. 9a, b for males; Supplementary Fig. 14a, b for females), while high $CO_2$ increased the fresh weight of 14-day-old thalli in both wild types and the mutants, probably due to enhanced photosynthesis. When serine was exogenously supplemented, suppressed growth of the mutants was recovered to some extent. However, the fresh weights of the mutants remained significantly lower than those of the wild type. The serine content in Mp*pgdh-1* under high $CO_2$ + serine conditions was approximately 70 nmol/mg DW, which was 3.5-fold higher than that in Tak-1 under high $CO_2$ conditions without serine supplementation (Supplementary Figure 15). Nevertheless, the Mp*pgdh-1* mutant under high $CO_2$ + serine conditions (approximately 100 mg FW, Fig. 9b) did not grow bigger than the wild type without serine supplementation (approximately 130 mg FW, Fig. 9b). This indicates that thallus growth did not depend only on total serine amount.

Reproductive growth was further observed under high $CO_2$. Wild types grew into maturity with healthy reproductive branches in about two months with far-red light, while both

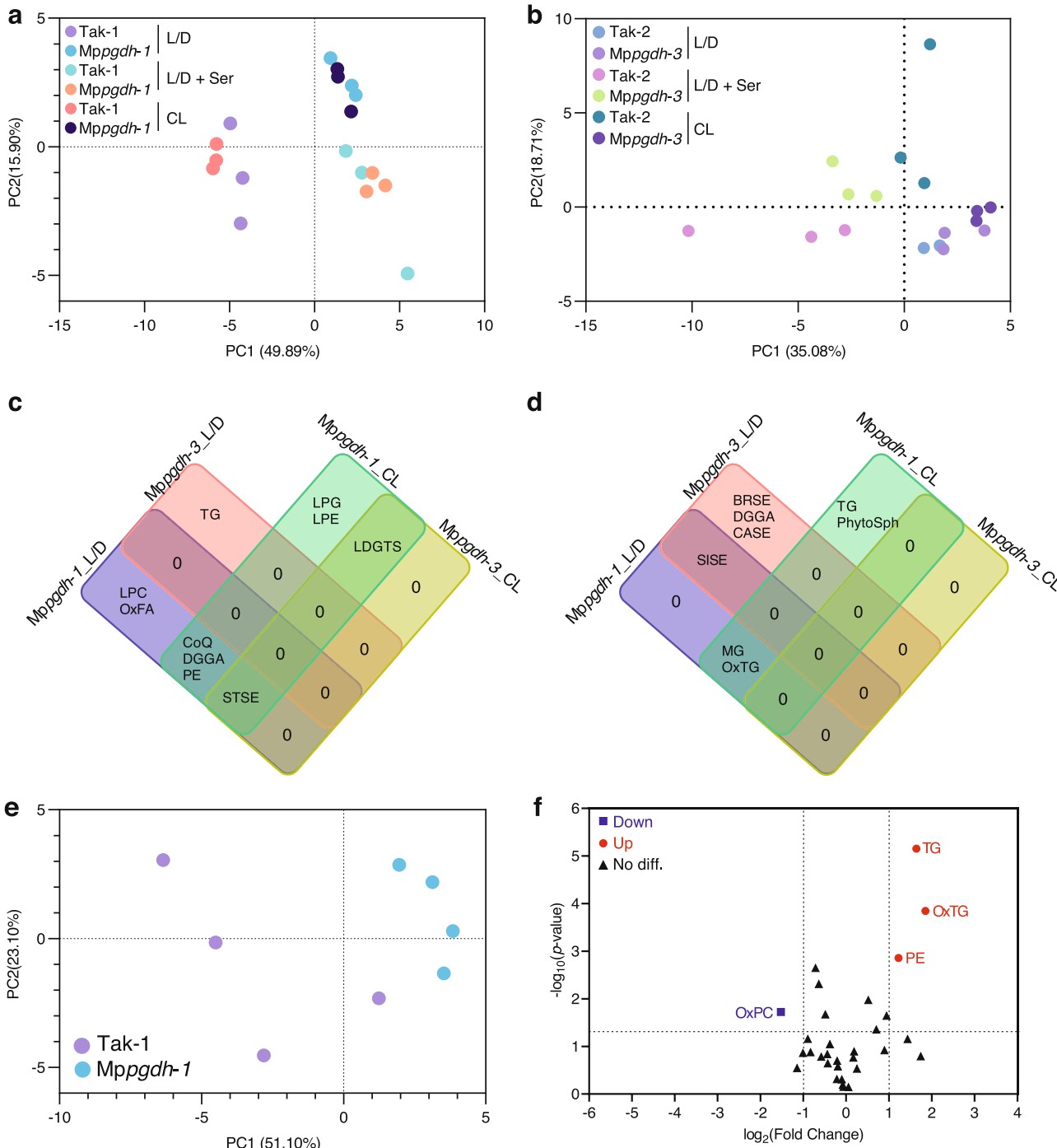

**Fig. 8 Changes in lipidome in the Mp*pgdh* mutants. a, b** PCA score plots of male (**a**) and female (**b**) 14-day thallus samples grown under L/D, L/D + serine, and CL conditions ($n = 3$). **c, d** Venn diagrams showing the significantly decreased (**c**) and increased (d) lipid classes in thalli of Mp*pgdh-1* and Mp*pgdh-3* under L/D and CL conditions. **e** PCA score plot of the antheridial receptacle samples grown under L/D conditions ($n = 4$). **f** Volcano plot showing the DALCs in antheridial receptacles of Mp*pgdh*-1. Red dots and blue squares represent significantly increased ($p$-value < 0.05, fold change > 2) and decreased ($p$-value < 0.05, fold change < 0.5) lipid classes, respectively, in Mp*pgdh*-1. Black triangles represent no significant differences between Tak-1 and Mp*pgdh*-1. TG triacylglycerol, OxTG oxidized triglyceride, LPG lysophosphatidylglycerol, LPC lysophophatidylcholine, PE phosphatidylethanolamine, LPE lysophosphatidylethanolamine, LDGTS lysodiacylglyceryl trimethylhomoserine/Lysodiacylglyceryl hydroxymethyl-*N,N,N*-trimethyl-β-alanine, OxFA oxidized fatty acid, CoQ coenzyme Q, DGGA diacylglyceryl glucuronide, STSE stigmasterol ester, SISE sitosterol ester, BRSE brassicasterol ester, CASE campesterol ester, MG monoacylglycerol, PhytoSph phytosphingosine, OxPC oxidized phosphatidylcholine.

male and female mutants were strongly impaired in growth and did not develop reproductive branches (Supplementary Fig. 16a). Interestingly, exogenous serine supplementation restored reproductive branch formation in Mp*pgdh* mutants (Supplementary Fig. 16a), while spermatogenesis was not fully restored

(Supplementary Fig. 16b). Considering that Mp*pgdh* mutants developed reproductive branches under L/D and ambient $CO_2$ conditions but failed in sperm development (Figs. 3a and 4a), the result indicated that enough serine supply from either the glycolate pathway or the PPSB is sufficient for reproductive

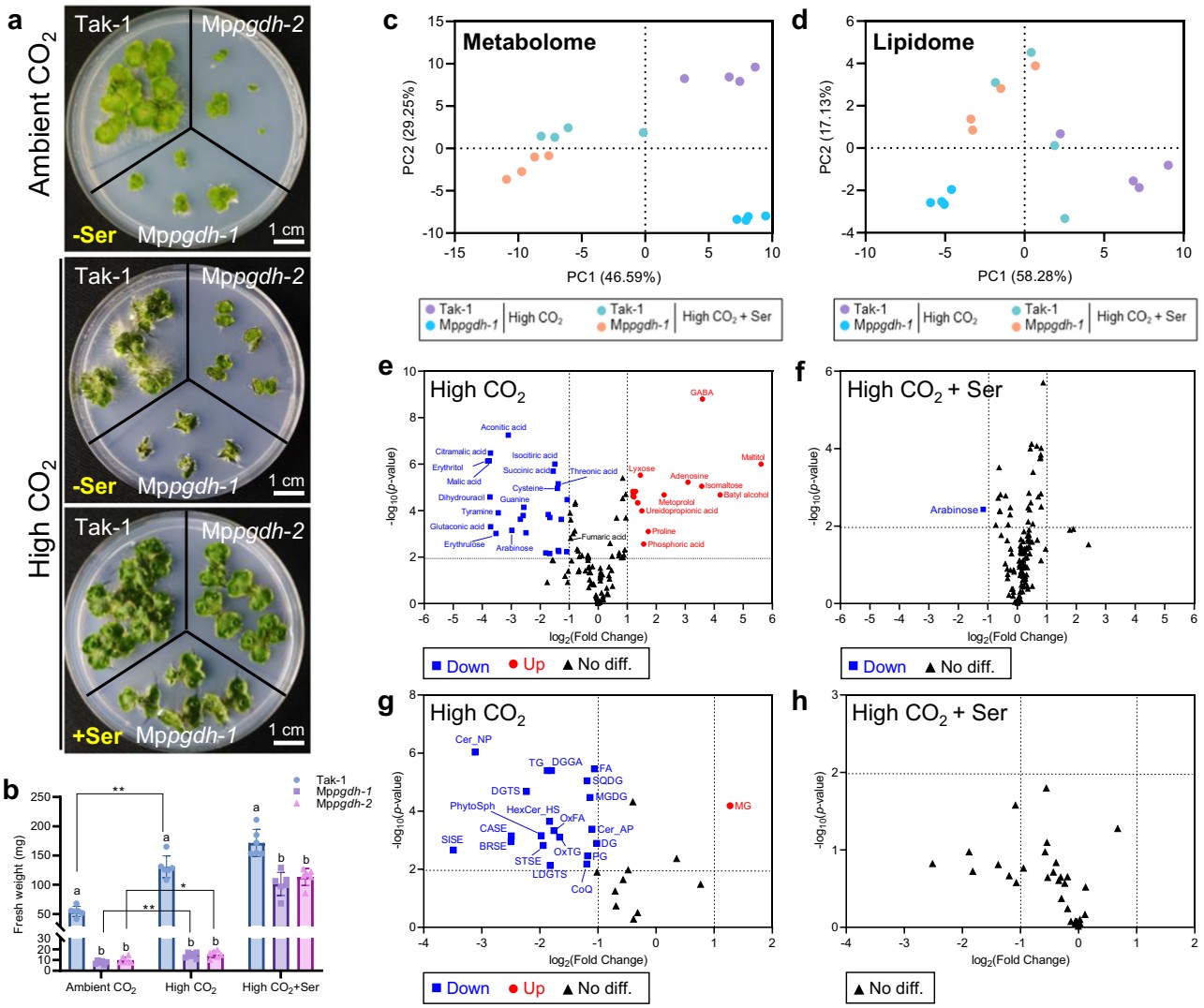

**Fig. 9 Growth and metabolic phenotypes of male Mp*pgdh* mutants under high $CO_2$ conditions. a** Plants grown on ½ B5 agar medium for 14 days under L/D conditions in ambient $CO_2$ (400 ppm) or high $CO_2$ (3000 ppm) with or without serine supplementation. **b** The fresh weight of Tak-1 and Mp*pgdh* mutants. Data represent means ± SD of six biological replicates ($n = 6$). One-way ANOVA followed by Tukey's test was performed ($p < 0.05$) in each growth condition; columns with the same letter indicate no significant differences. Student's *t*-test was performed in each line grown under ambient $CO_2$ and high $CO_2$ conditions. Asterisks indicate statistically significant differences (Student's *t*-test, *$p < 0.05$, **$p < 0.01$). **c, d** PCA score plots of the metabolome (**c**) and lipidome data (**d**) in 14-day-old thalli of Tak-1 and Mp*pgdh-1* grown under high $CO_2$ conditions with or without serine supplementation ($n = 4$). **e–h** Volcano plots showing DAMs (**e**, **f**) and DALCs (**g**, **h**) in Mp*pgdh-1* under the two growth conditions. Red dots and blue squares represent significantly increased ($p$-value < 0.01, fold change > 2) and decreased ($p$-value < 0.01, fold change < 0.5) metabolites or lipid classes, respectively, in Mp*pgdh-1*. Black triangles represent no significant differences between Tak-1 and Mp*pgdh-1*.

branch development, while the PPSB is necessary to normal spermatogenesis in *M. polymorpha*.

Metabolism in vegetative growth under high $CO_2$ was also investigated using metabolome and lipidome analyses. The PCA score plots (Fig. 9c, d) indicate that the metabolic and lipid profiles of Mp*pgdh-1* were apparently different from those of Tak-1 under high $CO_2$ conditions, and the difference between Mp*pgdh-1* and Tak-1 became smaller when serine was exogenously supplemented. The volcano plots (Fig. 9e–h) show that high $CO_2$ affected the accumulation of several metabolites and lipid classes in Mp*pgdh-1*, and such perturbed metabolic status was restored by exogenous serine supplementation. A similar trend was observed for Mp*pgdh-3* and Tak-2 (Supplementary Fig. 14c–h).

The metabolites and lipid classes whose accumulation were significantly affected by high $CO_2$ and Mp*PGDH* knockout were identified via Venn diagrams. A total of 15 DAMs decreased (Supplementary Fig. 17a), while 3 increased (Supplementary

Fig. 17b) commonly in male Mp*pgdh-1* and female Mp*pgdh-3*. The KEGG pathway analysis indicates that these DAMs were enriched ($p < 0.01$) in the metabolite sets "TCA cycle," "butanoate metabolism," "alanine, aspartate and glutamate metabolism" (Supplementary Fig. 17c). In the case of lipidome, 15 DALCs decreased in male and female mutants under high $CO_2$ conditions (Supplementary Fig. 17d), while no lipid class was commonly increased. Decreased DALCs included serine-derived lipid classes, such as ceramide alpha-hydroxy fatty acid-phytospingosine (Cer_AP), hexosylceramide hydroxyfatty acid-sphingosine (HexCer_HS).

## Discussion

In plants, among three serine biosynthesis pathways, the photorespiratory glycolate pathway is the primary producer of serine during the day[8]. Mp*PGDH* expression was induced when thallus

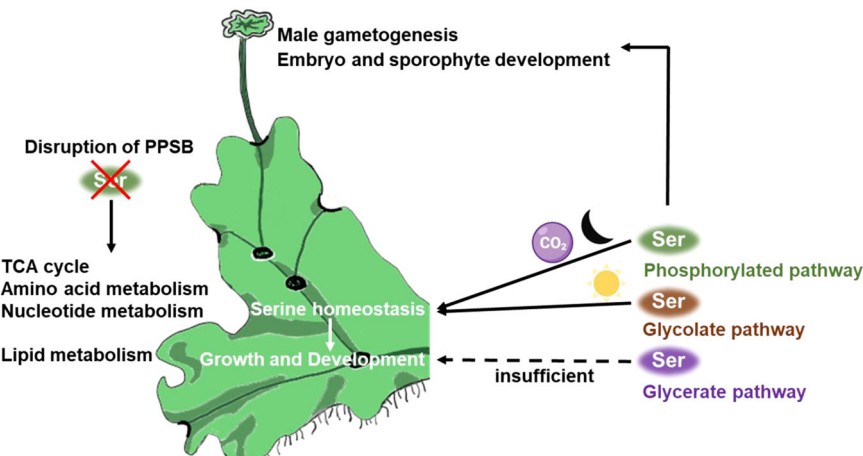

**Fig. 10 Proposed model of serine homeostasis in *M. polymorpha*.** In *M. polymorpha*, three pathways are involved in serine synthesis. The phosphorylated pathway is the primary serine synthesis pathway when the glycolate pathway is inactive in the dark or under high $CO_2$ conditions. The phosphorylated and glycolate pathways maintain stability in vivo serine homeostasis for normal growth and development. The phosphorylated pathway plays a unique role in male gametogenesis and embryo and sporophyte development. Disruption of the phosphorylated pathway of serine biosynthesis (PPSB) causes metabolic disorders. The existence and function of the glycerate pathway remain unclear (dashed line).

was transferred to darkness (Supplementary Fig. 5) and grown under high $CO_2$ conditions (Supplementary Fig. 2f), suggesting a mechanism to activate the PPSB in situations where serine synthesis by the glycolate pathway is suppressed. Consistent with this, the suppressed growth of Mp*pgdh* mutants was specifically observed under L/D conditions (Fig. 1 and Supplementary Fig. 4). Furthermore, the serine content in Mp*pgdh* mutants was strongly reduced during the dark period and rapidly restored to wild-type level under light (Fig. 2). This indicates that the decreased serine supply during the dark period caused poor growth of Mp*pgdh* mutants under L/D conditions. Under CL conditions, serine was continuously synthesized from the glycolate pathway, which was sufficient for growth of the Mp*pgdh* mutants (Fig. 1). These results further suggest that stable serine production throughout the day is necessary for vegetative growth.

The Mp*pgdh* mutants and the wild type grew better and appeared darker green under high $CO_2$ (3000 ppm) than under ambient $CO_2$ (Fig. 9a, b and Supplementary Fig. 14a, b), probably due to enhanced photosynthesis under high $CO_2$. This result was in contrast to a study using *A. thaliana*, in which elevated $CO_2$ (4000 ppm) improved the biomass of the wild type but not At*PGDH1* silencing lines[38]. The optimal $CO_2$ concentration for growth may vary depending on plant species and environmental factors such as nutrient availability and light intensity. Under ambient $CO_2$, the Mp*pgdh* mutants continued to grow very slowly, but eventually reached a size comparable to that of the wild type. On the other hand, under high $CO_2$, the mutant hardly grew further and exogenous serine supplementation did not promote thalli growth in the mutants as much as in the wild type. This indicated that serine is a limiting factor for growth under high $CO_2$ conditions, but metabolic homeostasis is crucial. Even wild type grew slower and produced fewer sperm under elevated $CO_2$ than under ambient $CO_2$ (Supplementary Fig. 16), suggesting high $CO_2$ widely affects metabolism and other limiting factors than serine for growth and development in *M. polymorpha* also exist under this condition.

Regarding gametogenesis, Mp*pgdh-1* barely produced sperm cells under L/D conditions (Fig. 3c, d). Most cells were arrested at the spermatid mother cell stage in the mutants (Fig. 3f). Mp*pgdh-1* produced a certain number of sperm cells when supplemented with serine or grown under CL conditions (Supplementary Fig. 7c–f). Zimmermann et al[38]. proposed that externally supplied

amino acids are metabolized similarly to those produced in plants. In our study, however, exogenously supplemented serine recuperated stagnant thallus growth but did not impaired spermatogenesis in Mp*pgdh-1*. One possibility is that serine supplied by the PPSB is essential for sperm formation. Another possible reason is that exogenously supplemented serine is not fully transported to the organ/tissue where serine is required. The long-distance transport of serine to the antheridial receptacles or serine translocation from the synthesis site toward the demanding antheridium might be insufficient to support normal developmental processes. It remains unclear whether gametophyte of *M. polymorpha* have photosynthesis-conducting cells[39]. Detailed spatiotemporal analysis of amino acid contents would provide insights into the importance of serine homeostasis in development. It is interesting to note that mutants in which spermatogenesis is arrested at the spermatid mother cell stage are hitherto unknown. Loss of Mp*DUO1*, which is expressed in spermatid mother cells, does not affect its division[31]. Further Mp*pgdh* analyses may provide hints for understanding cell division of spermatid mother cell to form spermatid.

Under L/D + high $CO_2$ conditions, wild types developed reproductive branches, while the growth of Mp*pgdh* mutants was arrested at a particular phase (Supplementary Fig. 16a). These findings suggest that serine supply only from the glycerate pathway, the possible third pathway of serine synthesis, is insufficient to maintain the sustained growth of *M. polymorpha*. To date, the existence of the glycerate pathway has been proven by genomic information; however, its function remains elusive. Further studies on this pathway will provide more information on serine homeostasis in plants.

Sexual reproduction is an important developmental process for plants throughout their life cycle. In *A. thaliana*, At*PSP1*- and At*PGDH1*-knockout lines showed drastic impairment of pollen development and were sterile[17–19], indicating the importance of PPSB in pollen development. In a dioecious plant *M. polymorpha*, although the male Mp*pgdh* mutants developed antheridiophores, sperm development was strongly impaired (Fig. 3a–d). After fertilization with wild-type sperm, sporophyte development on the archegoniophore of the female mutant stopped before differentiation along the apical-basal axis only under the L/D condition (Fig. 5 and Supplementary Fig. 9). As egg cell was normally produced (Fig. 4c) and fertilization occurred in the female

Mp*pgdh* mutant (Fig. 5d), internal serine supply seemed sufficient for these processes even without serine from PPSB. As the entire paternal genome is inactivated throughout embryogenesis in *M. polymorpha*[35], sporophyte from Mp*pgdh-3* x Tak-1 lacked functional PPSB. In liverworts, sporophyte is dependent on gametophyte for the supply of water and nutrients[39]. It is also possible that the PPSB in maternal gametophyte where sporophyte grows is essential for sporophyte development. Taken together, the knockout of MpPGDH affected male gametogenesis and embryo and sporophyte development. Serine from PPSB has importance in male and female plants differently during sexual reproduction, for gametogenesis in male and embryogenesis in female, in *M. polymorpha*.

Plants require a constant supply of serine for the biosynthesis of nucleic acids, amino acids, glutathione, and lipids[2,3]. PPSB is involved in ammonium and sulfur assimilation in *A. thaliana*;[17,38,40] therefore, it is reasonable to assume that a lack of serine from the PPSB would extensively affect metabolism in *M. polymorpha*. Our metabolome analysis identified the metabolites related to the tricarboxylic acid (TCA) cycle, such as aconitic acid, fumaric acid, and malic acid, as DAMs in Mp*pgdh* mutants (Fig. 6d, g). The result was consistent with the function reported in *A. thaliana* that PPSB represents a vital branching point in the metabolic flux between serine synthesis and the TCA cycle[3]. Among carbohydrates, the amounts of erythrulose and erythritol were greatly decreased in the Mp*pgdh* mutants (Fig. 6e; Supplementary Fig. 10b, c; Supplementary Fig. 11b). In yeast, these metabolites are derived from erythrose 4-phosphate in the pentose phosphate pathway[41]. Although the biosynthesis pathway of these metabolites in plants is still unclear, our result suggests that the PPSB affects central carbon metabolism. The amounts of proline and glutamine were increased in the dark in the mutants (Fig. 2). The amounts of glutaconic acid and 2-aminoadipic acid, the intermediates of lysine degradation, were reduced (Fig. 6b, e and Supplementary Fig. 17a). The DAMs detected in this study were enriched in amino acid metabolism, especially alanine, aspartate and glutamate metabolism (Fig. 6d, g and Supplementary Fig. 17c), indicating that lack of the PPSB-mediated serine widely affects metabolisms of various amino acids. Serine plays an indispensable role in the metabolism of one-carbon units[2]. One of the main outputs of one-carbon metabolism is nucleotide biosynthesis. The amount of cytidine, a component of RNA, was increased, whereas that of dihydrouracil, an intermediate in the catabolism of uracil, was reduced in both thalli and antheridial receptacles of Mp*pgdh*-1 (Fig. 6b, c and Fig. 7b). These detected DAMs suggest that serine from PPSB is related to nucleotide metabolism.

We clarified that the PPSB is strongly involved in lipid metabolism. Besides serving as structural components of membranes, lipids are involved as chemical messengers/signaling molecules in various developmental processes and responses to environmental stresses[42–46]. Lipid species vary greatly among different plants. Improved analytical technologies and bioinformatics in lipidomics[47] have enabled the investigation of highly complex composition of lipids in plants. In this study, we analyzed the lipid profile of *M. polymorpha* by using our cutting-edge lipidomics platform and discussed the link between serine metabolism and lipids. Under L/D and high $CO_2$ conditions, lipid profiles were changed in Mp*pgdh* mutants, and several DALCs were identified (Fig. 9g and Supplementary Fig. 14f). When serine was supplemented under these two conditions, the number of DALCs was greatly reduced (Fig. 9h and Supplementary Fig. 14h), indicating that lack of serine from the phosphorylated and the glycolate pathways caused great changes in lipid profiles. Two major serine-derived lipid classes are sphingolipids and phosphatidylethanolamines (PE). Sphingolipid biosynthesis starts

from the condensation of serine with palmitoyl-CoA[48], while PE biosynthesis starts with the conversion of serine to ethanolamine[49]. The contents of sphingolipids (phytosphingosine [PhytoSph], hexosylceramide hydroxyfatty acid-sphingosine [HexCer_HS], ceramide alpha-hydroxy fatty acid-phytosphingosine [Cer_AP], and ceramide non-hydroxyfatty acid-phytosphingosine [Cer_NP]) significantly decreased in Mp*pgdh* mutants only under high $CO_2$ conditions (Fig. 9g and Supplementary Fig. 14f). PE was also reduced in thalli of male mutant under ambient $CO_2$ but accumulated in antheridiophores (Fig. 8c, f). Overall, our findings indicate that TCA cycle, amino acid metabolism, and nucleotide metabolism were among the primary pathways affected by serine deficiency from the PPSB. This study also revealed that lipid metabolism was affected by perturbation of serine metabolism.

The coexistence of different serine synthesis pathways complicates our understanding of the role of serine in plant development and metabolism. This study explained the specific function of PPSB in *M. polymorpha* and explored the relationship between three serine synthesis pathways in plants. Here, we present a model for functions of serine supplied by different pathways and serine homeostasis in *M. polymorpha* (Fig.10). A large amount of serine from the photorespiratory glycolate pathway supports fundamental growth and development, and PPSB is the main pathway for serine supply at night when the glycolate pathway is inactive. Both pathways control serine homeostasis for normal growth and development. PPSB is essential for sperm and sporophyte development, and the glycolate pathway cannot fully compensate for the lack of PPSB. Serine from the glycolate pathway is sufficient for vegetative growth but not maintaining metabolic homeostasis. Lack of PPSB-derived serine triggers metabolic and lipidomic perturbation. Besides the TCA cycle, PPSB influences amino acid and nucleotide metabolism and affects lipid profiles. Serine supply solely from the glycerate pathway is insufficient for normal growth and development. Further research remains warranted to determine the possible existence of the glycerate pathway and clarify its specific function and relationship with the other two pathways. In conclusion, serine from different biosynthesis pathways has different functions in plants. This study demonstrated the importance of PPSB and serine homeostasis in growth and development.

## Methods

**Plant materials**. We used male strain Takaragaike-1 (Tak-1) and female strain Takaragaike-2 (Tak-2) as wild types. To generate Mp*pgdh* mutants using the CRISPR/Cas9 system, two gRNA sequences gRNA1 (5'-GCGCATCGTGCAAGACCGCG-3', targeting in the 1st exon) and gRNA2 (5'-TGAAGGCCGG-TAAGTGCGCT-3', targeting at the junction of 1st intron and 1st exon) were designed to target Mp*PGDH*. Next, the gRNA sequences were inserted into pMpGE_En03, and the gRNA expression cassette was transferred to pMpGE010[50] using LR Clonase.

To generate the lines carrying GUS reporter gene driven by the promoter of Mp*PGDH* (the *pro*Mp*PGDH*:*GUS* lines), the genomic sequence of the upstream region of Mp*PGDH* (5000 bp) was amplified with PCR using the genomic DNA of Tak-1 as a template. Subsequently, the PCR product was ligated into pMpGWB104[51] to construct a pMpGWB104:*pro*Mp*PGDH* plasmid.

All plasmids were purified using LaboPass™ Plasmid Mini. The vectors (pMpGE010-Mp*PGDH*-gRNA1, pMpGE010-Mp*PGDH*-gRNA2, and pMpGWB104:*pro*Mp*PGDH*) were introduced into sporelings produced by crossing Tak-1 and Tak-2 via

co-cultivation with an *Agrobacterium* strain harboring the vectors. The detailed transformation method is as previously described[52]. After screening independent $T_1$ plants following an antibiotic selection step, we obtained 3–5 candidate lines for each plasmid in the male and female backgrounds selected via PCR using rbm27-F/rbm27-R and rhf73-F/rhf73-R, respectively[53]. All primers used are listed in Supplementary Table 1.

**Growth conditions**. Wild-type and mutant *M. polymorpha* were cultivated on half-strength Gamborg's B5 medium (pH = 5.5–5.6) containing 1% agar with or without 10 mM serine at 22 °C. We used three chambers, CL with air, 16-h light/8-h dark with air, and 16-h light/8-h dark with 3000 ppm $CO_2$, to grow plants. To induce reproduction branches, we applied additional far-red light irradiation to each chamber. Nylon membranes were tiled onto the surface of the medium to ease the transfer of *M. polymorpha* from the agar medium.

Genetic crosses of *M. polymorpha* were performed under L/D and CL conditions as the video showed (https://www.youtube.com/watch?v=YFjfgr-wsy0). After male and female lines developed reproduction branches (approximately 1.5-month under L/D conditions, 1-month under CL conditions, and 2-month under L/D conditions in high $CO_2$), 50 μL of water was dropped on the surface of an antheridiophore (Stage 4) and left for 10 min. After pipetting five times, all water with sperm cells inside was taken and put to an immature archegoniophore. Post fertilization, 3–4 weeks were required to obtain mature sporangia that contain spores.

**Western blot analysis of PGDH**. Total protein from thalli of *M. polymorpha* was extracted in the sodium dodecyl sulfate (SDS) sample solution [2% (w/v) SDS, 62.5 mM Tris-HCl (pH = 6.8), 7.5% (v/v) glycerol, and 0.01% bromophenol blue]. After boiling for 5 min, the homogenate was centrifuged at $20,000 \times g$ for 10 min, and the resulting supernatant was used as the protein sample. The protein concentration was determined using a BCA protein assay (Pierce). Equal amounts of proteins were separated using SDS polyacrylamide gel electrophoresis, and then transferred onto a polyvinylidene difluoride membrane. A PGDH antibody prepared using the recombinant *Arabidopsis* PGDH1 protein as the antigen[54] was used at a 1:8000 dilution. Anti-Rabbit IgG, HRP-Linked F(ab')$_2$ Fragment Donkey (Cytiva) was used as secondary antibody at a dilution of 1: 100,000. Chemiluminescence was used for detection of horseradish peroxidase-conjugated secondary antibodies and visualized using LAS-3000 (Fujifilm). Uncropped blots are provided in Supplementary Fig. 18.

**Reverse transcription-quantitative PCR**. Total RNA was extracted from the thalli and antheridial receptacles using ISO-SPIN Plant RNA (NIPPON GENE). Next, total RNA (500 ng) was reverse-transcribed using ReverTra Ace™ qPCR RT Master Mix with gDNA Remover (TOYOBO). Quantitative real-time PCR was performed using THUNDERBIRD™ SYBR qPCR Mix (TOYOBO) in the StepOnePlus Real-Time PCR System (Applied Biosystems). The PCR cycling conditions were 95 °C for 15 min, 40 cycles of 95 °C for 15 s, and 60 °C for 60 s. The gene expression level was normalized to that of Mp*ACT1* to obtain a relative expression level.

The biological samples were analyzed in triplicate. Primers used for qPCR are listed in Supplementary Table 2.

**Hoechst staining of sperm cells**. Stage 4 antheridiophores were used to assess sperm production[36]. First, 50 μL water was dropped onto the surface of each antheridiophore and left for 10 min. After pipetting five times, 30 μL water was placed in a 1.5-mL

tube with 1.5 μL bisBenzimide H33342 trihydrochloride (1 mg/mL) (Sigma-Aldrich). After staining for 10 min, 10 μL solution was drawn to a C-Chip (NanoEnTek) to check fluorescence by confocal microscopy (OLYMPUS U-HGLGPS BX53).

**GUS histochemical assay**. The gemmae, thalli, antheridiophores, and archegoniophores of *pro*Mp*PGDH:GUS* lines were submerged in cold 90% acetone for at least 10 min. Next, the samples were washed twice with 50 mM sodium-phosphate buffer (pH = 7.0). Subsequently, the samples were transferred into 2 mL Eppendorf tubes or 25 mL centrifuge tube (VIOLAMP) with GUS staining buffer [50 mM Na-phosphate buffer (pH7.0), 0.1% Triton, 10 mM EDTA, 0.5 mM $K_3Fe(CN)_6$, 0.5 mM $K_4Fe(CN)_6$, 1 mM X-gluc (5-bromo-4-chloro-3-indolyl-β-D-glucuronide cyclohexylammonium salt)] and incubated at 37 °C under dark conditions (for approximately 16 h). Afterward, the stained samples were decolored and fixed in 70% ethanol overnight at 4 °C to remove chlorophyll and other plant pigments. Lastly, before observation under dissecting or light microscope (LEICA DFC450 C), the thalli were soaked in a solution [8 g choral hydrate: 3 mL water: 1 mL triglyceride] for 24 h to make them transparent.

**Electron microscopy of resin-embedded tissues and surface structure**. The samples were fixed with 4% paraformaldehyde and 2% glutaraldehyde in 50 mM sodium cacodylate buffer (pH = 7.4) for 2 h at 22 °C and overnight at 4 °C, then post-fixed with 1% osmium tetroxide in 50 mM cacodylate buffer for 3 h at room temperature. After dehydration in a graded methanol series (25, 50, 75, 90, and 100%), the samples were embedded in Epon812 resin (TAAB). Subsequently, ultrathin sections (100 nm) were cut using a diamond knife on an ultramicrotome (Leica EM UC7, Leica Microsystems, Germany) and placed on a glass slide. Afterward, the sections were stained with 0.4% uranyl acetate, followed by lead citrate solution, and coated with osmium using an osmium coater (HPC-1SW, Vacuum Device, Japan)[55]. The coated sections were observed using FE-SEM (SU8220, Hitachi High-Tech, Japan) with a yttrium aluminum garnet backscattered electron detector at an accelerating voltage of 5 kV.

The uncoated samples were observed and imaged with a low-vacuum tabletop SEM (Hitachi TM3000, Japan) using back-scattered electrons and an accelerating voltage of 15 kV.

**Preparation and observation of histological sections**. For the histological cross-sections, archegoniophore and antheridiophore were dissected and fixed overnight in formalin–acetic acid–alcohol (4% formalin, 5% acetic acid, 50% ethanol) at room temperature. Then, samples were dehydrated using a graded series of ethanol washes [50, 60, 70, 80, 90, and 95% (v/v); 60 min each] and stored overnight in 99.5% (v/v) ethanol at room temperature. Next, fixed specimens were embedded in Technovit resin (Kulzer), in accordance with the manufacturer's instructions, and sectioned using a microtome (RM2125 RTS; Leica Microsystems). Sections were stained with Toluidine Blue and photographed under a microscope (Leica DM6 B) connected to a CCD camera (DFC 7000 T; Leica Microsystems).

**Amino acid analysis by gas chromatography mass spectrometry**. Free amino acids were extracted from 4 mg of a freeze-dried powder sample in 1 mL of extraction solvent (methanol: Milli-Q water = 4:1) placed in 2 mL sampling tubes containing 3 mm zirconia beads using a tube rotator for 10 min. After centrifugation at 14,000 rpm at 22 °C for 10 min, 900 μL of supernatant was transferred to another tube. Next, the amino acids were extracted from the residues three more times, and the four

extracted solutions were mixed. The 25 μg [13]C-labeled amino acids (Cambridge Isotope Laboratories) in 3.6 mL extracted solution were derivatized using the EZ: faast[TM] for free amino acid analysis using GC-MS (Phenomenex). Afterward, 1 μL of the solution was injected using an AOC-20i auto-injector and subjected to amino acid analysis using GCMS-QP2010 Plus (Shimadzu Corporation). The absolute concentrations of the amino acids were calculated based on those of [13]C-labeled amino acids.

**Widely targeted metabolome analysis using gas chromatography–triple quadrupole mass spectrometry (GC-QqQ-MS).** For widely targeted metabolome analysis using GC-QqQ-MS, a 4 mg freeze-dried powder sample was mixed with 1 mL extraction solvent (80% methanol and 0.1% formic acid) in 2 mL sampling tubes containing 3 mm zirconia beads on Shake Master Neo (BMS, Tokyo, Japan). After centrifugation at 100,000 rpm at 22 °C for 1 min, 200 μL of supernatant was transferred to 1.5 mL tubes containing 200 μL extraction solvent and 20 μL adonitol (0.2 mg/mL). Three blank controls (extraction solvent) and 20 quality control (QC) samples (mixed test samples) were also prepared. After vortexing, the solvents in the tubes were evaporated using a rotary evaporator (Thermo Scientific, Savant, SPD121P, & UVS800DDA) for 3 h, following which 100 μL MOX reagent (2% methoxyamine·HCl in pyridine) was added to tubes to dissolve the samples at 30 °C and 1,200 rpm for 6 h. Subsequently, 50 μL MSTFA 1% TMCS (Thermo) was added to the tubes and incubated at 37 °C and 1200 rpm for 30 min. After centrifugation at 5000 rpm for 1 min, 50 μL supernatant was collected and analyzed using GC-QqQ-MS (AOC-5000 Plus with GCMS-TQ8040, Shimadzu Corporation). All samples were tested in random order. Two QC samples were injected at regular intervals (every six test samples) throughout the analytical run for continuous recalibration. Raw data were collected using GCMS solution software (Shimadzu Corporation). Lastly, quality-filtered metabolites were selected with signal-to-noise ratio >3 and QC relative standard deviation <30%. Calculation and normalization of peak area values were conducted using MRMPROBS and LOWESS/Spline normalization tools[56]. GC-MS/MS parameters[57] and MRM transitions[58] were used for widely targeted analysis.

**Lipidome analysis using liquid chromatography–quadrupole time-of-flight mass spectrometry (LC-QTOF-MS).** Five mg of freeze-dried powder samples were placed in a 2 mL centrifuge tube, mixed with an 800 μL of extraction solution (methyl tert-butyl ether/methanol = 3/1 (v/v) containing 1 μM of 1,2-didecanoyl-sn-glycero-3-phosphocholine, Sigma-Aldrich), and extracted by shaking at 900 rpm at 4 °C for 5 min on Shake Master Neo (BMS, Tokyo, Japan) using zirconia beads. Subsequently, 250 μL of water was added to the homogenate. After vigorous stirring on a vortex mixer and dark incubation for 15 min on ice, the homogenate was centrifuged at $1000 \times g$ for 5 min. Afterward, the 200 μL of upper layer was transferred to a new 1.5 mL microcentrifuge tube. Next, the organic phase was evaporated to dryness using a centrifugal concentrator (ThermoSavant SPD2010, Thermo Fisher Scientific) at room temperature. The residue was dissolved in 250 μL of ethanol and centrifuged at $10,000 \times g$ for 15 min. The supernatant was subjected to LC-MS/MS analysis[58]. The dataset was analyzed using MS-DIAL version 4.80[47]. The data processing parameters of minimum amplitude (for peak picking) and retention time tolerance (for peak alignment) were set to 100 and 0.1 min, respectively; however, the default parameters were used for the others. The annotation results were manually curated by considering the basis of the equivalent carbon number model of lipids, in which the elution behavior of molecules in reverse phase LC depends on the length of acyl chains and the number of double bonds in lipids[59]. The representative adduct form used for lipid quantification was determined by different polarity MS data. Lastly, the peak height was used for lipid quantification, and the total amount of each lipid class (including lipids with different acyl chains) was calculated and used for further analysis. Lipid abbreviations are listed in Supplementary Table 3.

**Statistics and reproducibility.** Volcano plots, Heatmaps, and KEGG pathway analysis were performed at MetaboAnalyst 5.0 (https://www.metaboanalyst.ca/home.xhtml). Venn diagrams were created by Webtools (https://bioinformatics.psb.ugent.be/webtools/Venn/). GC-QqQ-MS dataset was analyzed using MRMPROBS (http://prime.psc.riken.jp/compms/mrmprobs/main.html), and LC-QTOF-MS dataset was analyzed using MS-DIAL version 4.80 (http://prime.psc.riken.jp/compms/msdial/main.html). Principal component analysis and significance test were performed in GraphPad Prism9.0 (https://www.graphpad.com/). Comparisons between two groups were analyzed with unpaired $t$ tests. Comparisons between two groups or more were analyzed with one-way ANOVA, and significant main effects were followed up with Tukey's multiple comparisons tests. Data are reported as mean ± SD for column graphs and line charts. A $p$-value less than 0.05 or 0.01 was considered statistically significant. In volcano plots, $p < 0.01$ and fold change > 2 was set to identify increased metabolites, while $p < 0.01$ and fold change < 0.5 for decreased metabolites in mutants comparing with wild types. Data from this paper was collected from 3–11 biological replicates from each independent line. All figures were optimized by GraphPad Prism9.0. Exact details are listed in figure legends.

**Reporting summary.** Further information on research design is available in the Nature Portfolio Reporting Summary linked to this article.

## Data availability

The metabolome and lipidome data were deposited into DROP Met (http://prime.psc.riken.jp/menta.cgi/prime/drop_index) under accession number DM0052. Sequence data are available from the MarpolBase (https://marchantia.info/) under the following accession numbers: Mp*PGDH* (Mp8g16970), Mp*ACT1*(Mp6g10990), Mp*PSAT* (Mp1g15430), Mp*PSP* (Mp2g10500), Mp*SHMT* (Mp1g09830), Mp*GDH* (Mp2g07580), Mp*DUO1* (Mp1g13010), Mp*DAZ1* (Mp4g11380), Mp*MID* (Mp1g11830), Mp*RKD* (Mp3g04030), Mp*PRM* (Mp3g14390), Mp*TUA5* (Mp4g08430), Mp*LC7* (Mp6g01560), Mp*CEN1* (Mp1g00710), Mp*HMGBOX1* (Mp8g07450), Mp*HMGBOX2* (Mp2g12330), Mp*HMGBOX3* (Mp8g16760), Mp*HMGBOX4* (Mp2g04030), Mp*HMGBOX5* (Mp7g16870), Mp*TOP1* (Mp6g05370), Mp*TOP2* (Mp3g13420), Mp*TOP3α* (Mp1g02630), Mp*TOP3β* (Mp1g10070), Mp*ATG5* (Mp1g12840), Mp*ATG7* (Mp2g07850), Mp*ATG13* (Mp7g03210). The numerical source data underlying the graphs are provided as Supplementary Data 2.

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

## Acknowledgements

We greatly appreciate Drs. Mie Shimojima (Tokyo Institute of Technology) and Hiroyuki Imai (Konan University) for discussion on lipids, Mr. Kouji Takano (RIKEN Center for Sustainable Resource Science) for his help on preparing LC-MS/MS samples, Dr. Hideya Fukuzawa (Kyoto University) for his assistance with high CO₂ experiments, Ms. Yoriko Matsuda (Kyoto University) for her technical assistance with construction of CRISPR/Cas9 lines, Ms. Junko Takanobu (RIKEN Center for Sustainable Resource Science) for her help in preparing culture medium and weighing samples, and Dr. Takashi Araki

(Kyoto University) for his useful advice in interpretation of the results about male gametogenesis. We are grateful to Graduate Program of Transformative Chem-Bio Research (GTR) program in Nagoya University to support this study. This work was supported in part by JSPS KAKENHI Grant Numbers 25113010 and 20H04852, and GteX Program Japan Grant Number JPMJGX23B0 to M.Y.H.

## Author contributions

M.Y.H. conceived the idea for the project. M.Wang prepared the draft. M.Wang, H.Tabeta., and K.O. checked the growth and development phenotypes, including thalli growth, sperm production, and spore formation. M.Wang conducted GUS staining experiments and prepared samples for metabolic and lipidomic analysis. H.Tabeta. performed metabolic analysis and cross-section microscopic analyses. A.K. measured free amino acid contents. K.T., M.S., and M.Wakazaki performed the FE-SEM analyses. H.Tsugawa., T.S., and Y.O. performed lipidomic analysis. H.Tsugawa, T.I., M.Wang and M.Y.H. discussed lipidome annotation. H.A., T.K., R.N., and K.Y. supported the construction of Mp*pgdh* mutants and GUS lines. K.Y. performed western blot experiment. M.Wang, H.Tabeta., R.S., A.F., and M.Y.H. discussed the results and wrote the manuscript.

## Competing interests

The authors declare no competing interests.
