## [Peer Review File · Communications Biology]

Reviewers' comments:

Reviewer #1 (Remarks to the Author):

The function of the phosphorylated pathway of serine biosynthesis (PPSB) in a non-vascular plant has never been addressed. This manuscript nicely describes the characterization of PGDH, the rate-limiting enzyme of the PPSB in the model bryophyte species *Marchantia polymorpha*.

This work confirms findings found in higher plants, such as the crucial role of PPSB in vegetative growth, or its requirement for sperm formation or sporophyte development (embryo development in higher plants), even though the species are so far apart phylogenetically. This would confirm the essentiality of PPSB in the plant kingdom.

The introduction is clear, concise, and updated and the results were rigorously carried out. The discussion could be improved. I have some comments that may help to improve the manuscript.

Comments.

1) In the introduction it is mentioned that the non-photorespiratory pathways are involved in serine biosynthesis in dark environments. This statement should be clarified, since the PPSB may also be involved in serine biosynthesis during the day as stated later in the introduction, results, and discussion, to maintain metabolic homeostasis and for tissue and organ development, at least for specific cell types. In agreement with my comment, the authors show in the results section that the metabolomes of WT and PPSB mutants are different even under continuous light conditions which confirms the role of PPSB during the day.

2) The authors found that the PPSB mutants show retarded growth under light/dark conditions but not under continuous light conditions. However, to determine whether the poor growth of the mutant thalli was due to insufficient endogenous serine supply, they measured the amino acid content during the light period and did not find differences in serine content in one of the mutants. Why does one mutant show serine deficiency while the other does not show it? These points should be clarified.

Furthermore, in my opinion, in order to see serine insufficiency, it would have been more convenient to measure the serine content during the dark period. In fact, the authors demonstrated that exogenously added serine could revert the retarded growth phenotype, which suggests a serine deficiency in the mutants. It is also possible that there is no serine insufficiency in the mutants at the whole plant level. It may be that the serine deficiency is restricted to specific cell types involved in growth. All these considerations should be addressed.

3) The result section "DNA -dependent processes..." is too long and difficult to read. It could be considerably shortened by eliminating all negative results.

4) Supplementary Table S3 is missing.

5) Line 334. The subheading states "PPSB is the primary serine synthesis pathway when photorespiration is inhibited". Later on, the authors show that under high CO₂ conditions, both wild-type and mutants grow better than under ambient CO₂. This is surprising since mutants are lacking serine biosynthesis from both PPSB and glycolate pathways, which in my opinion would limit growth (at least this is what happens in *Arabidopsis* PPSB deficient lines). Does this mean that serine is not a limiting factor for growth in *Marchantia*? Then, why PGDH mutants are smaller than WT? The authors should provide a reasonable explanation for all these apparent contradictions.

6) The discussion should also be improved taking in consideration the comments raised above.

Minor comments.

Line 214. Change Figure 3 to Figure 4.

Line 390. Supplementary Fig. 2Si does not refer to MpPGDH expression. Wrong figure.

Reviewer #2 (Remarks to the Author):

The manuscript presented by Wang et al. describes the requirement of phosphorylated serine for *Marchantia* growth and development. The authors provide good evidence for the expression profile of the MpPGDH enzyme and they also provide detailed phenotypic characterization of Mppgdh mutant vegetative and reproductive development specifically when grown under long-day conditions. Transcript profiling by qRT-PCR and metabolomics/lipidomics profiling further characterizes the contribution of the phosphorylated serine pathway to liverwort growth and development. I'm supportive of the manuscript overall but there a couple of issues that need to be addressed.

1. I'm somewhat confused by the result that Mppgdh-1 does not have a defect in endogenous serine. This would argue that serine isn't the primary cause for the observed phenotype. Can the authors provide any explanation on this key issue? Is it more appropriate to be measuring metabolites specific to the phosphorylated serine pathway directly? This point should be clarified in line 147 to keep the reader on track.

Discussion lines 394-397: since serine supply is often referred to throughout the paper I think it is important to expand on this section. If it isn't serine per se that is impacted, what is it? This is critical to all interpretations of the work.

2. Since the main focus of this manuscript is the phenotypic characterization of Mppgdh mutants, having data to demonstrate genetic complementation, even if only performed for growth phenotypes, would be an appropriate/important control to include.

Minor points:

Line 98: It might be worth adding 'in vitro' here to clarify that your previous study was focused on the function of purified proteins. This may provide a clearer distinction when introducing your new study.

In fig 2d: this should be expressed as a standardized value. Sperm cells per antheridia appears to be more appropriate when comparing two genotypes.

Figure S5: This is a useful piece of data but it misses a CL control that was not moved to dark. This would be useful to show that changes are due to environmental factors and not part of the circadian rhythm. Instead of performing additional experiments, I'm wondering if the authors can take advantage of existing circadian rhythm expression datasets (perhaps this set: <https://nph.onlinelibrary.wiley.com/doi/full/10.1111/nph.17653>) to address this potential issue.

The term 'reproduction branches' appears to be inconsistent with the field, which often refers to these as gametophores. Is this what the authors are referring to here? Also, the images in Fig S16a are fairly low resolution.

Reviewer #3 (Remarks to the Author):

The authors present an interesting study on functions of serine from the phosphorylated pathway on growth, male gametogenesis, and metabolism in liverworts. There is a comprehensive dataset on genetics, physiology and omics. The authors prepared high quality figures and the manuscript is well written. I only have a few minor comments.

1. In the title, "Functions of serine from the phosphorylated pathway" is a little vague. The authors are suggested to modify the title that will highlight the major findings of the paper.
2. Keywords are missing from the manuscript. The authors may need to add some based on the requirement of the journal.
3. Introduction: Pages 3 and 4: There are lots of text on the PGDH in Arabidopsis. The author need to add relevant studies on PGDHs in other important species such as rice (there are three OsPGDHs) and *Physcomitrium patens* (there are four PpPGDHs).
4. Add an overarching hypothesis to the last paragraph of the Introduction
5. Can the authors please explain the significant growth phenotype of Mppgdh-2 in terms of low Fresh Weight in Figure 2?
6. Figure 5: please add the full gene names in the Figure legend.
7. Figure 8: there are only small number of differentially regulated lipids in the Mppgdh mutants. The authors should add full name of those lipids shown in the Venn diagrams (c,d) in the Figure legend.
7. Figures 8 and 9 should be combined into one large figure.
8. Figure 10: The high CO₂ experiment is one of the highlights of the paper. The authors should justify the use of 3000ppm of CO₂ for 14 days. What is the optimal CO₂ level for liverworts from an evolutionary point of view? The plants were grown on media with no nutrient limitation. However, the high CO₂ + serine treatment may have a different impact if the plants are grown in pots or in the natural environment where mineral nutrients are limiting. The authors may want to discuss this as well.

We greatly appreciate the valuable comments and suggestions provided by the editor and the reviewers. We have performed additional experiments and addressed the discussion points as recommended by the reviewers, although only genetic complementation experiments have not been completed due to technical reasons (please refer to one-by-one responses below). While revising the manuscript, we discussed with an expert regarding the interpretation of the results related to the defect in sperm development observed in *Mppgdh* mutants. Accordingly, the main text has been slightly revised (highlighted in red in the revised manuscript). In the original version, the Abstract and the main text exceeded the character limit, leading us to condense them without altering the meaning.

The following are our one-by-one responses to the reviewers' comments.

Reviewer #1 (Remarks to the Author):

*The function of the phosphorylated pathway of serine biosynthesis (PPSB) in a non-vascular plant has never been addressed. This manuscript nicely describes the characterization of PGDH, the rate-limiting enzyme of the PPSB in the model bryophyte species *Marchantia polymorpha*.*

This work confirms findings found in higher plants, such as the crucial role of PPSB in vegetative growth, or its requirement for sperm formation or sporophyte development (embryo development in higher plants), even though the species are so far apart phylogenetically. This would confirm the essentiality of PPSB in the plant kingdom.

The introduction is clear, concise, and updated and the results were rigorously carried out. The discussion could be improved. I have some comments that may help to improve the manuscript.

Comments.

1) In the introduction it is mentioned that the non-photorespiratory pathways are involved in serine biosynthesis in dark environments. This statement should be clarified, since the PPSB may also be involved in serine biosynthesis during the day as stated later in the introduction, results, and discussion, to maintain metabolic homeostasis and for tissue and organ development, at least for specific cell types. In agreement with my comment, the authors show in the results section that the metabolomes of WT and PPSB mutants are different even under continuous light conditions which confirms the role of PPSB during the day.

Thank you very much for your comments. In Introduction and Results, we edited the corresponding sentences as follows to explain clearly that non-photorespiratory pathways are also involved in serine biosynthesis in the daytime.

In Introduction L49-55 ‘In plants, three serine biosynthesis pathways exist and function coordinately in the daytime. The major pathway during daytime is the photorespiratory glycolate pathway in photosynthetic tissues (Supplementary Fig. 1). …… In dark environments and non-photosynthetic tissues, two other pathways are responsible for serine synthesis, namely, the glycerate and phosphorylated pathways.’

In Result L230-232 ‘In the PCA score plot, *Mppgdh-1* formed the clusters separated from those of Tak-1 not only under L/D condition but also under CL condition, indicating that the PPSB also works at daytime.’

2) The authors found that the PPSB mutants show retarded growth under light/dark conditions but not under continuous light conditions. However, to determine whether the poor growth of the mutant thalli was due to insufficient endogenous serine supply, they measured the amino acid content during the light period and did not find differences in serine content in one of the mutants. Why does one mutant show serine deficiency while the other does not show it? These points should be clarified. Furthermore, in my opinion, in order to see serine insufficiency, it would have been more convenient to measure the serine content during the dark period. In fact, the authors demonstrated that exogenously added serine could revert the retarded growth phenotype, which suggests a serine deficiency in the mutants. It is also possible that there is no serine insufficiency in the mutants at the whole plant level. It may be that the serine deficiency is restricted to specific cell types involved in growth. All these considerations should be addressed.

Thank you very much for your comments. We performed additional experiments to sample thalli during the dark period and analyze amino acid contents. The result was shown in Fig. 2 in the revised manuscript. After 4 h in the dark, the serine amount in *Mppgdh* mutants was reduced significantly at the whole plant level compared with wild types. After 2 h in the light period, the serine content in *Mppgdh* mutants was recovered to wild type-level. We added the following sentences in Results and Discussion.

L136-139: ‘The serine content in *Mppgdh* mutants was significantly decreased after 4 h in the dark and restored to wild type-level after 2 h in the light (Fig. 2). These results suggested that reduced

serine content during the dark period caused the poor growth in *Mppgdh* mutants under L/D conditions.'

L345-348: 'Furthermore, the serine content in *Mppgdh* mutants was strongly reduced during the dark period and rapidly restored to wild-type level under light (Fig. 2). This indicates that the decreased serine supply during the dark period caused poor growth of *Mppgdh* mutants under L/D conditions.'

Fig. 2 Free amino acid contents in thalli of the *Mppgdh* mutants under L/D conditions.

The thalli of male lines grown under 16 h light/8 h dark conditions were sampled at 0 h, 2 h, 4 h, 6 h, and 8 h in the dark period, and at 2 h and 14 h in the light period. The free amino acid contents were measured using gas chromatography–quadrupole mass spectrometry. Data represent means \pm SD of three biological replicates. Student's *t*-test was performed between wild type and one mutant respectively. Asterisks indicate statistically significant differences between the wild type and both two mutant lines (* p < 0.05, ** p < 0.01).

3) *The result section “DNA -dependent processes...” is too long and difficult to read. It could be considerably shortened by eliminating all negative results.*

During revising the manuscript, we discussed the qRT-PCR result with Prof. Takashi Araki in Kyoto University, whose expertise is sperm development in *Marchantia polymorpha*. According to your comments and Prof. Araki's advice, we deleted the section “DNA-dependent processes are impaired during gametogenesis in male *Mppgdh* mutants” and modified the section “MpPGDH-mediated serine synthesis is essential to sperm formation” as follows.

L162-172: ‘However, in the *Mppgdh* mutants, spermatid mother cells were in various sizes and irregular shapes (Fig.3f, bottom left), and subsequent diagonal cell division to generate spermatids was barely observed (Fig. 3f, bottom middle and right). The qRT-PCR analysis of sperm differentiation-related genes (Supplementary Fig. 8) revealed that the expression of *MpPRM*, which is expressed specifically in sperm during late spermiogenesis³⁷, were significantly repressed. On the other hand, the expression level of *MpDUO1*, which is a key transcription factor of *MpPRM* and expressed in spermatid mother cells and spermatids³⁶, was similar in the wild type and the mutants. Additionally, a member in *MpHMGBOX* gene family, *MpHMGBOX4* was also strongly repressed in the *Mppgdh* mutants. These results support the notion that cell division of spermatid mother cells to generate spermatid was blocked in the *Mppgdh* mutants.’

4) *Supplementary Table S3 is missing.*

Supplementary Table S3 was submitted in Excel form and was not merged with other supplementary data in PDF form.

5) *Line 334. The subheading states “PPSB is the primary serine synthesis pathway when photorespiration is inhibited”. Later on, the authors show that under high CO2 conditions, both wild-type and mutants grow better than under ambient CO2. This is surprising since mutants are lacking serine biosynthesis from both PPSB and glycolate pathways, which in my opinion would limit growth (at least this is what happens in Arabidopsis PPSB deficient lines). Does this mean that serine is not a limiting factor for growth in Marchantia? Then, why PGDH mutants are smaller than WT? The authors should provide a reasonable explanation for all these apparent contradictions.*

Under high CO₂, wild-type and mutant thalli became darker green, thicker with a rough surface, and the number of rhizoids increased compared with under ambient CO₂. We consider that the increase in the fresh weight of 2-week-old wild-type and *Mppgdh* mutants was caused by enhanced

photosynthesis (Fig. 9a). However, when we grew the plants for a prolonged time, wild type took more time to develop reproductive branches than under ambient CO₂. In the case of *Mppgdh* mutants, thalli did hardly grow further under high CO₂, but exogenous serine supplementation can induce the development of reproductive branches (Supplementary Fig. 16a). These results suggest serine is a limiting factor for growth and development of *Marchantia*. However, because 3000 ppm CO₂ widely affects metabolism, other limiting factors must exist under this condition. Based on the comments from you and Reviewer 3, we discussed the phenotypes under high CO₂ conditions as follows.

L352-366: ‘The *Mppgdh* mutants and the wild type grew better and appeared darker green under high CO₂ (3000 ppm) than under ambient CO₂ (Fig. 9a, b and Supplementary Fig. 14a, b), probably due to enhanced photosynthesis under high CO₂. This result was in contrast to a study using *A. thaliana*, in which elevated CO₂ (4000 ppm) improved the biomass of the wild type but not *AtPGDHI* silencing lines³⁹. The optimal CO₂ concentration for growth may vary depending on plant species and environmental factors such as nutrient availability and light intensity. Under ambient CO₂, the *Mppgdh* mutants continued to grow very slowly, but eventually reached a size comparable to that of the wild type. On the other hand, under high CO₂, the mutant hardly grew further and exogenous serine supplementation did not promote thalli growth in the mutants as much as in the wild type. This indicated that serine is a limiting factor for growth under high CO₂ conditions, but metabolic homeostasis is crucial. Even wild type grew slower and produced fewer sperm under elevated CO₂ than under ambient CO₂ (Supplementary Fig. 16), suggesting high CO₂ widely affects metabolism and other limiting factors than serine for growth and development in *M. polymorpha* also exist under this condition.’

6) The discussion should also be improved taking in consideration the comments raised above.

We have revised the discussion based on your comments. Changes are highlighted in red in the revised manuscript.

Minor comments.

Line 214. Change Figure 3 to Figure 4.

We are sorry for the mistakes in citing figures. In the revised manuscript, a new figure (Fig.2 Free amino acid contents in thalli of the *Mppgdh* mutants under L/D conditions) has been added and a main figure was moved to Supplementary Figures (Supplementary Fig. 8 The expressions of the

spermatogenesis-related genes in antheridial receptacles). Accordingly, the figure numbers were changed.

Line 390. Supplementary Fig. 2Si does not refer to MpPGDH expression. Wrong figure.

Supplementary Fig. 2Si (Supplementary Fig. 2f in the revised manuscript) showed the GUS-staining image of *proMpPGDH:GUS* lines under high CO₂ conditions, where thalli became thicker and harder and were not easy to stain. We are very sorry that the quality of photo is not good, but most part of thalli was stained blue, while only midrib was stained under ambient CO₂ condition (Supplementary Fig. 2c).

Reviewer #2 (Remarks to the Author):

The manuscript presented by Wang et al. describes the requirement of phosphorylated serine for Marchantia growth and development. The authors provide good evidence for the expression profile of the MpPGDH enzyme and they also provide detailed phenotypic characterization of Mppgdh mutant vegetative and reproductive development specifically when grown under long-day conditions. Transcript profiling by qRT-PCR and metabolomics/lipidomics profiling further characterizes the contribution of the phosphorylated serine pathway to liverwort growth and development. I'm supportive of the manuscript overall but there a couple of issues that need to be addressed.

1. I'm somewhat confused by the result that Mppgdh-1 does not have a defect in endogenous serine. This would argue that serine isn't the primary cause for the observed phenotype. Can the authors provide any explanation on this key issue? Is it more appropriate to be measuring metabolites specific to the phosphorylated serine pathway directly? This point should be clarified in line 147 to keep the reader on track.

Discussion lines 394-397: since serine supply is often referred to throughout the paper I think it is important to expand on this section. If it isn't serine per se that is impacted, what is it? This is critical to all interpretations of the work.

Thank you very much for your comment. Previously, we analyzed serine content of thalli sampled during the light period. In the revised manuscript, we performed additional experiments to sample thalli during the dark period and showed time-dependent change in serine contents in new Fig. 2. After 4 h in the dark, the serine content in Mppgdh mutants was significantly reduced compared with

the wild type, and after 2 h in the light period, the serine content in *Mppgdh* mutants was recovered to wild type-level. Based on the result, we edited the text throughout the manuscript. Changes are highlighted in red in the revised manuscript.

2. Since the main focus of this manuscript is the phenotypic characterization of Mppgdh mutants, having data to demonstrate genetic complementation, even if only performed for growth phenotypes, would be an appropriate/important control to include.

We thank you for the comment and agree with you. We designed the target site of gRNA2 at the junction between exon and intron (Supplementary Fig. 3a) to perform genetic complementation experiment. We constructed the vector *proMpPGDH::MpPGDH* in pMpGWB301 and we have been trying transformation of two *Mppgdh* mutants and wild type Tak-1(control) with this vector. However, the selection marker (chlorsulfuron resistance) of pMpGWB301 does not work well in our experimental conditions. Regeneration of all lines including wild type on selection medium was not observed, so we have not yet obtained transformant at this moment. On the other hand, in the original manuscript we used two and three alleles (Supplementary Fig. 3) for male and female mutants, respectively, showing similar growth phenotypes. In addition, we demonstrated by the additional experiment (Fig. 2) that serine contents in the mutants were significantly reduced in the dark, indicating that knockout of *MpPGDH* gene resulted in the phenotypic changes. Considering that many *Marchantia* studies were published without genetic complementation experiment (although it is not desirable situation), we decided to resubmit the manuscript without genetic complementation result before the resubmission deadline. We sincerely apologize for this.

Minor points:

3. Line 98: It might be worth adding 'in vitro' here to clarify that your previous study was focused on the function of purified proteins. This may provide a clearer distinction when introducing your new study.

Thank you very much for your suggestion. We have added 'in vitro' and 'in vivo' to the following sentences to clarify the distinction between our previous study and this study.

L91-95: 'MpPGDH has similar biochemical characteristics to AtPGDH1 *in vitro*, such as cooperative inhibition by L-serine and activation by L-alanine, L-valine, L-methionine, L-homoserine, and L-homocysteine. In this study, we aim to clarify the *in vivo* function of *MpPGDH* and explore the specific functions of serine from the phosphorylated pathway in a non-vascular plant *M. polymorpha*.'

4. In fig 2d: this should be expressed as a standardized value. Sperm cells per antheridia appears to be more appropriate when comparing two genotypes.

We apologize that the label of Fig. 2d was not appropriate. The value was standardized by the volume of water dropped and the label should be ‘Number of sperm cell/uL’. We agree with you that the unit ‘Sperm cells per antheridia’ is more appropriate because probably the mutants have fewer antheridia than wild type.

Besides, during the revision, we realized that the nuclei of sperm were very thin and crescent-shaped in the photo of Hoechst staining. However, we had counted fluorescent spots as sperm cells without carefully checking their shape. In the photos, the fluorescent spots (sometimes cell masses) may contain other cell types such as spermatogonia, and it was not appropriate to count the number of fluorescent spots. Therefore, we removed the bar graphs in Fig. 2d, Supplementary Fig. 8, and Supplementary Fig. 16 in the original manuscript. Nevertheless, it was apparent that the number of fluorescent cells including sperm was reduced in the mutants and the conclusion remains unaffected.

5. Figure S5: This is a useful piece of data but it misses a CL control that was not moved to dark. This would be useful to show that changes are due to environmental factors and not part of the circadian rhythm. Instead of performing additional experiments, I'm wondering if the authors can take advantage of existing circadian rhythm expression datasets (perhaps this set: <https://nph.onlinelibrary.wiley.com/doi/full/10.1111/nph.17653>) to address this potential issue.

We sincerely appreciate your suggestion. We agreed that a CL control is useful and performed additional experiments to add CL control data in Supplementary Fig. 5. The expression of genes was relatively stable under CL conditions. This result clearly showed that changes in gene expression levels are due to environmental factors and not part of the circadian rhythm. Accordingly, we add a phrase “compared to CL conditions” in L130.

Supplementary Fig. 5 Expression of serine biosynthesis pathway genes in the dark.

The 14-day-old Tak-1 thalli grown under the CL condition were transferred to the dark condition or continued to grow under light condition. Total RNA was extracted from Tak-1 at 0, 4, 8, 16, and 24 h after transfer. Two thalli were used as one sample. The expressions of MpPGDH, MpPSAT, MpPSP, MpGDH, and MpSHMT were determined by qRT-PCR. The gene expression was normalized against that of MpACT1.

6. The term ‘reproduction branches’ appears to be inconsistent with the field, which often refers to these as gametophores. Is this what the authors are referring to here? Also, the images in Fig S16a are fairly low resolution.

In this manuscript, we collectively refer to male antheridiophores and female archegoniophores as ‘reproductive branches.’

We apologize for the low resolution of Supplementary Fig. 16a. In this experiment, *Marchantia polymorpha* was grown in a bottle for around 2 months. The thalli climbed along the wall of the bottle, and the reproductive branches also extended upward unevenly. As a result, it was difficult to focus when taking photos, and we could only focus on one plane. The purpose of Supplementary Fig. 16a is to show the wild types developed reproductive branches under high CO₂ conditions, while *Mppgdh* mutants did not. Although the resolution of the images is not so high, we think they show the difference between wild types and mutants.

Reviewer #3 (Remarks to the Author):

The authors present an interesting study on functions of serine from the phosphorylated pathway on growth, male gametogenesis, and metabolism in liverworts. There is a comprehensive dataset on genetics, physiology and omics. The authors prepared high quality figures and the manuscript is well written. I only have a few minor comments.

1. In the title, "Functions of serine from the phosphorylated pathway" is a little vague. The authors are suggested to modify the title that will highlight the major findings of the paper.

Thank you very much for your comment. We have modified the title to ‘The phosphorylated pathway of serine biosynthesis affects sperm, embryo, and sporophyte development, and metabolism in *Marchantia polymorpha*’.

2. Keywords are missing from the manuscript. The authors may need to add some based on the requirement of the journal.

Thank you very much for your comment. After carefully reading the submission guidelines for *Communications Biology*, it appears that including keywords is not an explicit requirement. However, based on your suggestion, we would like to list the following keywords: Phosphorylated pathway of serine biosynthesis, 3-Phosphoglycerate dehydrogenase, Male gametogenesis, Metabolism, Serine homeostasis’.

3. Introduction: Pages 3 and 4: There are lots of text on the PGDH in Arabidopsis. The authors need to add relevant studies on PGDHs in other important species such as rice (there are three OsPGDHs) and Physcomitrium patens (there are four PpPGDHs).

We have shortened the text on *Arabidopsis* and added relevant studies on PGDHs in other plant species in Introduction as follows.

L72-82: ‘In *A. thaliana*, three PGDH isoforms (*AtPGDH1*, *AtPGDH2*, and *AtPGDH3*) are expressed in different organs/tissues and have different physiological functions. *AtPGDH1* is the essential gene and its silencing causes developmental arrest in roots, embryos, pollen, and male gametophytes. *AtPGDH2* has a partially redundant role with *AtPGDH1*, while *AtPGDH3* seems to have additional functions unrelated to serine synthesis^{17-19,22,23}. In rice (*Oryza sativa*), three *OsPGDH* genes were expressed in all tissues and development stages, and their expressions responded to abiotic stresses²⁴. Additionally, four *PpPGDHs* and two *AmtriPGDHs* were identified in moss *Physcomitrium patens*

and basal angiosperm *Amborella trichopoda*, respectively²⁵. The biochemical properties of various PGDH enzymes are different among isozymes in terms of allosteric regulation by amino acids^{25,26}.’

4. Add an overarching hypothesis to the last paragraph of the Introduction.

We have added an overarching hypothesis to the last paragraph of Introduction as follows.

L101-102: ‘This study proposes that serine homeostasis is a key factor for robustness of not only metabolism but also growth and development.’

5. Can the authors please explain the significant growth phenotype of Mppgdh-2 in terms of low Fresh Weight in Figure 2?

Growth of *Mppgdh* mutants is delayed but not totally impaired. In other words, growth and development of *Mppgdh* mutants were slower but finally reached wild-type level. Additionally, *Mppgdh-2* always exhibits more severe phenotypes than *Mppgdh-1*. In Figure 2, the fresh weight of antheridiophores of *Mppgdh-2* was lower than Tak-1 and *Mppgdh-1*, because Tak-1 and the mutants were sampled at the same time. However, if *Mppgdh-2* was grown in the chamber for a longer time, the fresh weight could reach a wild-type level.

6. Figure 5: please add the full gene names in the Figure legend.

In the revised manuscript, Figure 5 has been moved to Supplementary Fig. 8. Full gene names have been added to the legend.

7. Figure 8: there are only small number of differentially regulated lipids in the Mppgdh mutants. The authors should add full name of those lipids shown in the Venn diagrams (c, d) in the Figure legend.

We have added the full lipid names in figure legend as follows:

‘TG, triacylglycerol; OxTG, oxidized triglyceride; LPG, lysophosphatidylglycerol; LPC, lysophosphatidylcholine; PE, phosphatidylethanolamine; LPE, lysophosphatidylethanolamine; LDGTS, lysodiacylglyceryl trimethylhomoserine/Lysodiacylglyceryl hydroxymethyl-*N,N,N*-trimethyl- β -alanine; OxFA, oxidized fatty acid; CoQ, coenzyme Q; DGGA, diacylglyceryl glucuronide; STSE, stigmaterol ester; SISE, sitosterol ester; BRSE, brassicasterol ester; CASE, campesterol ester; MG, monoacylglycerol; PhytoSph, phytosphingosine; OxPC, oxidized phosphatidylcholine.’

8. *Figures 8 and 9 should be combined into one large figure.*

Figures 8 and 9 were combined into one large figure as ‘Fig. 8 Changes in lipidome in the *Mppgdh* mutants.’

9. *Figure 10: The high CO₂ experiment is one of the highlights of the paper. The authors should justify the use of 3000 ppm of CO₂ for 14 days. What is the optimal CO₂ level for liverworts from an evolutionary point of view? The plants were grown on media with no nutrient limitation. However, the high CO₂ + serine treatment may have a different impact if the plants are grown in pots or in the natural environment where mineral nutrients are limiting. The authors may want to discuss this as well.*

Thank you very much for giving us an interesting point of view. Because *Marchantia polymorpha* is a basal land plant, it is reasonable to consider that *Marchantia polymorpha* has optimal CO₂ level which differs from those of *Arabidopsis thaliana* and other angiosperm. It is reported that different plant species have different optimal CO₂ concentrations. For example, Lee *et al.* (<https://doi.org/10.1046/j.1469-8137.2001.00095.x>) and Wang *et al.* (<https://doi.org/10.1007/s00442-010-1572-x>) showed a substantial increase of the biomass of some plants when CO₂ concentration increased from about 350 ppm to 700 ppm, but the CO₂ fertilization effect on plant growth declined or vanished beyond certain CO₂ concentrations (such as 1000 ppm). In this study, we used the 3000 ppm CO₂ condition based on other studies (for example, Benstein *et al.* (2013) *Plant Cell*; <https://academic.oup.com/plcell/article/25/12/5011/6098476>). We apologize for not being able to provide the optimal CO₂ level for *Marchantia* without checking its growth under various CO₂ concentrations.

Previous studies demonstrated that the PPSB is associated with nitrogen and sulfur metabolism. Furthermore, in this study, abundant serine supplementation had a significant impact on entire metabolism probably by perturbing carbon, nitrogen, and sulfur metabolism. We completely agree with your view that the high CO₂ + serine treatment may have a different impact if the plants are grown in pots or in the natural environment where mineral nutrients are limiting. According to the comments from you and Reviewer 1, we modified the discussion as follows.

L352-366: ‘The *Mppgdh* mutants and the wild type grew better and appeared darker green under high CO₂ (3000 ppm) than under ambient CO₂ (Fig. 9a, b and Supplementary Fig. 14a, b), probably due to enhanced photosynthesis under high CO₂. This result was in contrast to a study using *A. thaliana*, in

which elevated CO₂ (4000 ppm) improved the biomass of the wild type but not *AtPGDH1* silencing lines³⁹. The optimal CO₂ concentration for growth may vary depending on plant species and environmental factors such as nutrient availability and light intensity. Under ambient CO₂, the *Mppgdh* mutants continued to grow very slowly, but eventually reached a size comparable to that of the wild type. On the other hand, under high CO₂, the mutant hardly grew further and exogenous serine supplementation did not promote thalli growth in the mutants as much as in the wild type. This indicated that serine is a limiting factor for growth under high CO₂ conditions, but metabolic homeostasis is crucial. Even wild type grew slower and produced fewer sperm under elevated CO₂ than under ambient CO₂ (Supplementary Fig. 16), suggesting high CO₂ widely affects metabolism and other limiting factors than serine for growth and development in *M. polymorpha* also exist under this condition.'

REVIEWERS' COMMENTS:

Reviewer #1 (Remarks to the Author):

The authors have addressed all my suggestions and comments. They have also performed additional experiments which I believe has improved their manuscript. I am fully satisfied with their responses.

Reviewer #2 (Remarks to the Author):

Where feasible the authors addressed all of my main concerns. Happy to support publication.

Reviewer #3 (Remarks to the Author):

The authors addressed the comments properly. Thanks!